# Machine learning-enabled forward prediction and inverse design of 4D-printed active plates

Xiaohao Sun [1,6], Liang Yue [1,6], Luxia Yu[1,6], Connor T. Forte[1], Connor D. Armstrong[1], Kun Zhou [2], Frédéric Demoly [3,4], Ruike Renee Zhao [5] & H. Jerry Qi [1] ✉

Shape transformations of active composites (ACs) depend on the spatial distribution of constituent materials. Voxel-level complex material distributions can be encoded by 3D printing, offering enormous freedom for possible shape-change 4D-printed ACs. However, efficiently designing the material distribution to achieve desired 3D shape changes is significantly challenging yet greatly needed. Here, we present an approach that combines machine learning (ML) with both gradient-descent (GD) and evolutionary algorithm (EA) to design AC plates with 3D shape changes. A residual network ML model is developed for the forward shape prediction. A global-subdomain design strategy with ML-GD and ML-EA is then used for the inverse material-distribution design. For a variety of numerically generated target shapes, both ML-GD and ML-EA demonstrate high efficiency. By further combining ML-EA with a normal distance-based loss function, optimized designs are achieved for multiple irregular target shapes. Our approach thus provides a highly efficient tool for the design of 4D-printed active composites.

Active composites (ACs), consisting of active materials that respond differently to external stimuli, can transform their shapes once stimulated. Typical stimuli include heat[1–4], light[5,6], water[7,8], and magnetic field[9–11]. The active shape change depends on the spatial distributions of constituent active and passive materials, see the simplest AC bilayer for example (Fig. 1a). Multimaterial 3D (or 4D) printing technology[12–15] is ideal to encode unique material distribution in a highly voxelized AC to generate pre-programmed, non-intuitive shape changes, which provides great design freedom. Fully leveraging the large design space and manufacturing flexibility of 4D printing requires solving a challenging design problem, i.e., efficiently finding the optimal material distribution to achieve a target shape change[16,17]. To address the design problem for 4D printing, computational design strategies including gradient-based and gradient-free methods have been developed. For

example, the gradient-based topology optimization (TO) has been used to guide the design for 4D printing, such as designing shape-changing behaviors of ACs[18–20] or optimizing compliances of soft actuators[21]. The TO, however, typically requires the complicated derivation of gradients and may encounter difficulties when the active material involves geometric or material nonlinearity (e.g., multiphysics-driven material nonlinearity). Alternatively, the gradient-free method, such as finite element (FE)-based evolutionary algorithm (EA), has achieved great success in designing certain shape-change responses of active composites[22–24] or other engineering structural problems[25–28]. Such FE-EA approach generally relies on numerous FE calculations to explore a large design space, thus suffering from high computational cost. To resolve this issue, efforts have been made on developing the reduced-order forward models which enables more

[1]The George W. Woodruff School of Mechanical Engineering, Georgia Institute of Technology, Atlanta, GA, USA. [2]Singapore Centre for 3D Printing, School of Mechanical and Aerospace Engineering, Nanyang Technological University, Singapore, Singapore. [3]ICB UMR 6303 CNRS, Belfort-Montbeliard University of Technology, UTBM, Belfort, France. [4]Institut universitaire de France (IUF), Paris, France. [5]Department of Mechanical Engineering, Stanford University, Stanford, CA, USA. [6]These authors contributed equally: Xiaohao Sun, Liang Yue, Luxia Yu. ✉e-mail: qih@me.gatech.edu

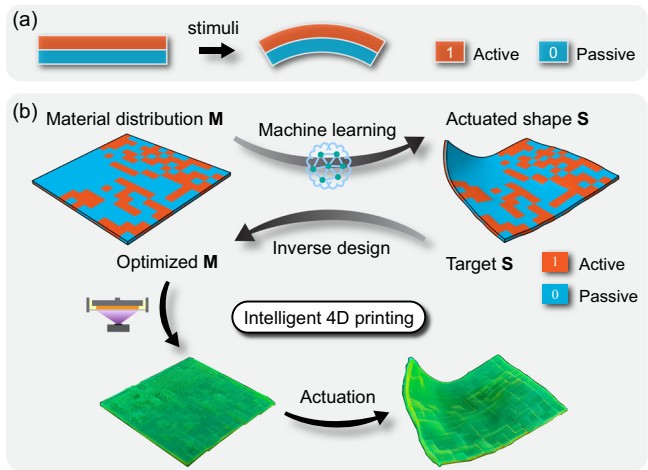

**Fig. 1 | Overview of machine-learning (ML)-enabled, voxel-level inverse design of 4D-printed active composite (AC) plates. a** Example of AC actuation: bending of a bilayer due to mismatch in the active response (eigenstrain) of constituent active and passive materials. **b** Schematic of a complete design-fabrication process for AC plates: the forward prediction of actuated shape through ML, the inverse design of material distribution through ML-incorporated optimization, and the 4D-printed realization of the voxel-level optimized design. Two materials are encoded as '1' (active) and '0' (passive). The voxel-level material distribution is then digitally encoded into a three-dimensional binary number array (denoted by **M**). The actuated shape is parameterized as coordinate data of sampling points (denoted by **S**). For forward prediction, the ML model takes an input **M** and produces the output **S**. For inverse design, the ML-based optimization algorithm receives a target **S** and produces the optimized **M**.

efficient inverse design[29,30]. Yet, accurately and efficiently exploring a large design space and tackling inverse problems remain to be challenging.

Recent advancements in machine learning (ML)[31] offer new possibilities for developing fast, computationally affordable, and high-fidelity predictive models that can be integrated with the optimization algorithm to achieve an efficient inverse design. For example, Gu and coworkers have made extensive explorations[32–35] on utilizing ML capabilities, such as combining ML with gradient descent (GD) and active learning[32] or with EA[33], for materials design. However, existing works mainly focused on optimizing mechanical properties of materials, such as strength and toughness of composites[34,35], auxetic metamaterials[36,37], and responses of soft pneumatic robots[38]; there is limited work on ML-based design of shape changes of ACs[39–43]. Compared to optimizing or extremizing a few properties, the design for shape changes has unique challenges such as highly complex mapping from material distributions to shapes (particularly for large deflections), the high-dimensional nature of the shape data and the variability of the target shapes. This also places higher demands on the accuracy of ML models. Therefore, there exist big gaps between existing methodologies and the shape-change design of 4D-printed ACs. For AC beams, Zhang et al.[39] utilized ML approach to solve the forward prediction problem. Our recent work[40] presents an ML-EA approach that demonstrates high efficiency in the inverse design of AC beams with complicated target shape change, which cannot be achieved by conventional methods such as FE-EA. The recurrent neural network (RNN) is proven to be appropriate to deal with sequential data arising from the beam deflection[41]. However, the active beam design is essentially a one-dimensional (1D) structure with 2D shape change, which has relatively small design space, is more manageable but has limited applications. The inverse design of 2D active structures (e.g., active plates) with 3D deformation is highly desired to further leverage the large design space of 4D printing and ACs, which could have much broader applications. However, this is also significantly more

challenging due to the higher-dimensional and more intricate deformation behaviors, the more complicated data mapping, and the tremendously increased design space. For example, as discussed later, for an active beam design using $24 \times 4$ voxels for 2D deformation, the design space is ($2^{96} \approx 8 \times 10^{28}$); for an active plate with $15 \times 15 \times 2$ voxels, the design space becomes $2^{450}$ ($\approx 3 \times 10^{135}$), an increase of more than $10^{100}$. Dealing with such a tremendous design space requires new strategies.

It should be noted that shape transformations of 2D AC sheets into 3D surfaces due to differential expansion have long been studied to understand biological morphogenesis[44] or harnessed to program shape changes[45,46]. These studies have focused on certain geometries, such as spherical or saddle surfaces. More recently, great progress has been made in the inverse design for arbitrary target surfaces. One important approach is to exploit established techniques of conformal mapping to obtain a spatially varying, isotropic expansion field, or a metric, that maps the planar surface to the target[47]. The expansion can be physically realized through not only material expansion such as swelling, plasticity and growth[48,49], but also specially designed mechanical units such as auxetic materials[50,51] and others[52]. Note that the metric uniquely determines the Gaussian curvature ($K$) of the surface, and thus the conformal mapping is essentially finding an optimal metric to achieve target $K$. In fact, the metric change is not limited to the isotropic expansion. Other geometric mapping strategies have been proposed to optimize the metric with certain constraints for various anisotropic physical systems[53–55]. Although achieving great success, the geometric approach suffered from some limitations. First, the Gaussian curvature alone does not determine a unique state but can adopt different isometric configurations[47,48]. For example, a developable, arbitrarily bent surface is isometric to the planar state ($K = 0$) and thus cannot be achieved through the geometric approach. In this case, voxel-level designs that allow material heterogeneities in the thickness direction are important. Second, mechanics has not been accurately considered in the geometric approach. In general, the shapes prescribed by a designed metric represent stretch-free configurations[56]. To achieve so, this approach essentially minimizes stretching energy without considering bending energy, and the shapes are correct only when the bending energy is negligible compared to the stretching energy, e.g., infinitely thin sheet. Thus, in many practical systems where bending is non-negligible, the geometric design can lead to the shape that deviates significantly from the target and thus is often used as the initial solution for further optimizations with mechanics models[52,53]. In addition to the geometric mapping, various strategies for surface designs have been developed for specific physical systems, such as nematic sheets[55,57], origami[58,59], lattices[60,61], buckling mesosurfaces[62], and inflatable systems[63,64]. However, a universal and efficient voxel-level design approach is highly desired, particularly for 4D printing, as it can enable an integrated and intelligent design-fabrication process.

This work tackles the challenging problem of designing active plates, i.e., exploits ML for the efficient forward shape prediction and inverse design for voxelized plates, as illustrated in Fig. 1b. The volumetric expansion is used to represent a general active response (eigenstrain) that can be induced by different mechanisms. A deep residual network (ResNet)-based ML model is trained using a dataset acquired from FE simulations. By combining key ingredients such as the data augmentation by symmetries and the proper selection of boundary conditions, the trained ML model achieves high accuracy in the shape prediction with complex material distributions. We then incorporate the ML into both the GD and the EA (gradient-free) algorithms for finding the optimal material distribution based on a target shape. The ML forward model accelerates the gradient-based optimization by enabling not only the fast forward prediction but also the efficient computation of exact gradients via automatic differentiation (AD). In addition, the ML allows the EA to search solutions in a large

design space that is impossible to be explored by FE simulations. Furthermore, using a global-subdomain design strategy with the two algorithms, optimized designs in terms of material distributions for a variety of FE-derived and algorithmically generated target shapes (or surfaces) are rapidly achieved. By further combining ML-EA with a normal distance-based loss, optimized designs are achieved for multiple irregular target shapes. Although many ML-based strategies have been utilized for the materials design[32–35], applying existing methodologies to the shape-change design of 4D-printed ACs is very challenging due to the high complexities outlined above. Our voxel-level inverse design approach empowered by ML pave the way for the intelligent design and fabrication of 4D printing, and is thus promising in facilitating broader applications of 4D-printed shape-morphing AC structures.

## Results

### Physical problem and dataset

In this work, we assume the active plate consists of two materials, a material with an active expansion (referred to as active material) upon the stimulus and a material that does not respond to the stimulus (referred to as passive material), which are encoded as "1" and "0", respectively, as shown in Fig. 1b. The two materials are randomly assigned to $N_x \times N_y \times N_z$ voxels, with $N_x$ and $N_y$ being the voxel numbers in two in-plane (x- and y-) directions and $N_z$ in the thickness (z-) direction, respectively. The material distribution is therefore digitally encoded into a 3D binary number array of "1"s and "0"s, denoted by $\mathbf{M}$. Under a certain stimulus, the active material expands while the passive material does not, resulting in a shape change of the plate (Fig. 1b). The deformed shape is represented by the coordinates $(x_i, y_i, z_i)$ of voxel mesh points sampled on the mid-surface, giving a 3D number array denoted by $\mathbf{S}$. For the studied active plate, there exists $2^{N_x \times N_y \times N_z}$ possible material distributions, giving a large and complex design space. For example, here, we consider the case with voxel numbers of $15 \times 15 \times 2$ (with the dimension of 40 mm × 40 mm × 1 mm), which has a huge design space of $2^{450}$ ($\approx 3 \times 10^{135}$) and is much greater than that of the active beam considered in our previous work ($2^{96} \approx 8 \times 10^{28}$)[40].

The FE model for the active plate is developed to generate the dataset (see "Methods"). We propose four boundary conditions (BCs) that allow the AC plate to deform freely, which mimic the active response of most ACs in the literature. More

specifically, we adopt the following BCs in our FE simulations

$$\begin{aligned} x_A = y_A = z_A = 0, \\ x_C = y_C, z_C = 0, \\ z_B = z_D. \end{aligned} \quad (1)$$

where $A, B, C, D$ are four corner points of the mid-surface of the plate as illustrated in Fig. 2. The first equation of Eq. (1) fix $A$ at the origin and eliminate the translation of the plate, and the other three equations eliminate the rotation at $A$ about the $z$ axis, about the axis $(-1,1,0)$ and about the axis $(1,1,0)$, respectively. Once we have simulation data, we can easily convert the deformed shapes to those satisfying different BCs that allow free deformation. Here, we propose such a set of BCs to mimic the scenario where one corner of the plate is clamped as follows,

$$\begin{aligned} x_A = y_A = z_A = 0, \\ x_{C'} = y_{C'}, z_{C'} = 0, \\ z_{B'} = z_{D'}. \end{aligned} \quad (2)$$

where $B', C', D'$ are the three corner points (or nodes) of the corner element at the mid-surface (Fig. 2). Note that Eq. (2) follows similar forms with Eq. (1) but is prescribed at different points, which enables the free deformation of the plate but mimic a *corner-clamped* condition. We will refer to the BCs in Eqs. (1) and (2) as the original and converted BCs, respectively. In addition, the other two BCs (BC3 and BC4), which are detailed in Supplementary Note 1, will also be evaluated. As will be shown in "Performance of machine-learning model", the converted BCs are most beneficial for the ML prediction and thus will be used throughout the paper unless otherwise specified.

Due to the large design space, a large amount of training data is needed. We create 56,250 random material distributions (or designs) of two types: 31,250 fully random designs and 25,000 island designs. The former is randomly generated without any pattern constraints, while the latter is obtained through certain combinations of random island images with connected domains of "1"s or "0"s (see "Methods" and Supplementary Fig. 1). The corresponding true actuated shapes are then obtained through FE simulations (Fig. 2). As a result, the generated datasets are pairs of material distribution $\mathbf{M}$ and actuated shape $\mathbf{S}$ in terms of coordinates of sampling points. Owing to the symmetries in the geometry, each simulation data can be augmented

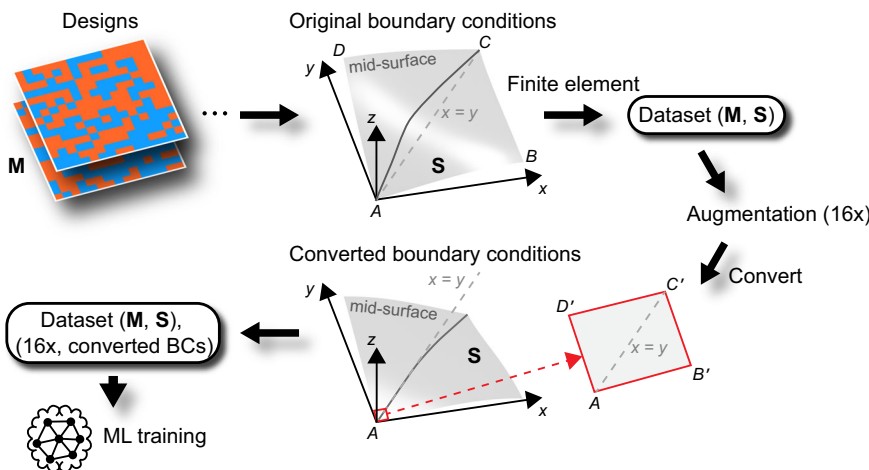

**Fig. 2 | Schematics of two boundary conditions (BCs): original BCs (Eq. (1)) and converted BCs (Eq. (2)).** $\mathbf{M}$ and $\mathbf{S}$ represent the random material distribution designs and corresponding actuated shapes, respectively. Using the original BCs, FE simulations are performed to generate the dataset, i.e., pairs of ($\mathbf{M}$, $\mathbf{S}$). The same BCs are then used to augment the dataset 16 times. Finally, the converted BCs are used to transform the generated dataset for the ML training. The quadrilateral AB'C'D' (red) is taken to be the corner element face that lies on the mid-surface.

to 16 non-repeating data with flipped and/or transposed material distribution designs (Supplementary Fig. 2), whose actuated shapes can be calculated using symmetric rules without need for new simulations. As the original BCs allow for simple calculation of the new shape changes, the dataset is augmented first based on the original BCs and then converted to the other set of BCs. After 56,250 FE simulations and data augmentation, the size of the entire dataset is 900,000, which is split into training and validation datasets with fractions of 0.9 and 0.1, respectively. As detailed in "Methods", statistics investigations of the dataset are performed, and the results show that the island dataset overall exhibits larger displacements (mean value of maximum displacement: 33.0 mm) than that of the fully random dataset (mean value of maximum displacement: 8.4 mm) (Supplementary Fig. 3a, b); the training and validation datasets follow similar distributions (Supplementary Fig. 3c). The data is then used to train the ML model, which will be constructed below.

## Machine-learning model for forward prediction

In this section, we present a ResNet-based ML model that can accurately predict the actuated deformation of an AC plate. The ResNet architecture is well suited for very deep convolutional neural networks (CNNs) and has achieved widespread successes in various computer vision tasks, such as image classification[65]. ResNet has also been used to predict the mechanical response of magneto-mechanical materials[43]. To illustrate the essential idea, we show in Fig. 3a the ResNet used, which is comprised of $N$ residual blocks. Each block consists of two convolutional layers (each followed by a batch normalization (BN) layer and a rectified linear unit (ReLU) activation layer), an addition layer, and an identity skip connection (Fig. 3b). The block input is thus allowed to bypass the layers of the main branch, providing a simpler path through the block. For the entire ResNet, the inclusion of many skip connections allows information to propagate more easily between shallow and deep layers, thus mitigating issues of vanishing gradients and degradation in deep CNNs. In addition to the ResNet, we also build the plain CNN with the same architecture as ResNet but without skip connections, as well as the graph convolutional network (GCN), and compare their performances on the active plate design problem.

We use $\mathcal{F}$ to denote the built ML model, which receives a material distribution $\mathbf{M}$ and predicts the actuated shape $\mathbf{S}$ (or $(\mathbf{x}, \mathbf{y}, \mathbf{z})$), i.e.,

$$\mathbf{S} = (\mathbf{x}, \mathbf{y}, \mathbf{z}) = \mathcal{F}(\boldsymbol{\theta}; \mathbf{M}) \tag{3}$$

where $\mathbf{x}$, $\mathbf{y}$, or $\mathbf{z}$ consist of values of the specific coordinate component of all sampling points, $\boldsymbol{\theta}$ denotes the parameter set (weights and biases) of $\mathcal{F}$. The loss function is then defined as the sum of squared errors (SSE) between the predicted response $(\mathbf{x}, \mathbf{y}, \mathbf{z})$ and the true response $(\mathbf{x}^{true}, \mathbf{y}^{true}, \mathbf{z}^{true})$ for a specific design, i.e.,

$$\mathcal{L}(\boldsymbol{\theta}; \mathbf{M}) = \left\| (\mathbf{x}, \mathbf{y}, \mathbf{z}) - (\mathbf{x}^{true}, \mathbf{y}^{true}, \mathbf{z}^{true}) \right\|^2, \tag{4}$$

where $\|\cdot\|$ denotes the $L_2$ norm operator. The training of the network is to find the optimal $\boldsymbol{\theta}$ that minimizes the loss Eq. (4) on the training set, i.e.,

$$\min_{\boldsymbol{\theta}} \mathcal{L}(\boldsymbol{\theta}; \mathbf{M}) \tag{5}$$

The actual training process is performed over minibatches of the training set, i.e., by iteratively calculating the mean loss over a minibatch and making an update of $\boldsymbol{\theta}$. The Adam, a gradient-descent based optimizer, is used for the parameter updates. More details on the construction and training of the ML model are described in "Methods".

## Performance of machine-learning model

We first study how the four BCS, the original BCs (Eq. (1)), converted BCs (Eq. (2)), BC3 and BC4, affect the ML performance. Two ResNet models with different depths are trained using the data with the four BCs and their training curves are shown in Fig. 3c. The ResNet-7 (7 represents the number of convolutional layers), which has limited capability for learning complex shapes, achieves similar performance on all the BCs, while the ResNet-33 demonstrates much better performance on the converted BCs compared to the other three BCs. This suggests that shapes with the converted BCs are easier to learn by the deep ML models. The reason can be explained as follows. Physically, the converted BCs endow the deformed shape (in terms of coordinates) with some spatially sequential dependency on the voxel, starting from the fixed boundary towards the free boundary of the plate. In contrast, the original BCs result in a shape where each sampling point's coordinate depends on the entire voxel information, implying a more nonlinear mapping between the design and the shape. Therefore, we will use the ML model with the converted BCs throughout the paper.

Next, we study the effects of network architecture and depth on the model performance. Figure 3d shows the training curves of validation loss versus epochs for the plain CNN and ResNet with different numbers of convolutional layers. The training loss is quantitively similar to the validation loss and is thus omitted for visual clarity. The final validation loss values at epoch 20 for these cases are given in Fig. 3e. The results show that deeper network architectures generally lead to better performance, but plain CNN encounters degradation when the number of convolutional layers exceeds 21, which is not observed in ResNet. As a result, ResNet outperforms plain CNN for very deep networks, achieving its best performance for the number of convolutional layers equal to 51. For deeper ResNet, the bottleneck design is often adopted[65], which does not improves the performance in our case (Supplementary Fig. 4). Therefore, we will not use deeper network but the ResNet-51 in the rest of this paper. In addition, we also study the performance of GCN models, which proves to be inferior to that of both plain CNN and ResNet (Supplementary Fig. 5).

Inspired by our previous work[40], we use different networks to learn the individual coordinates (i.e., $x$, $y$, and $z$) so as to further improve the model performance. In particular, we consider the following three combinations of the coordinates for the training: (1) one network to learn $(x, y, z)$, (2) three networks to learn $(x)$, $(y)$, and $(z)$, respectively, and (3) two networks to learn $(x, y)$ and $(z)$, respectively. For each network of each case, the loss function takes a form similar to Eq. (4), with only slight changes in the specific coordinates. The total loss of different networks of each combination is then used to make the comparison. Figure 3f shows that the third combination, i.e., one for $(x, y)$ and one for $(z)$, demonstrates the best performance.

Based on the results above, we train the ML model with the ResNet-51 architecture, the converted BCs, and the combinations of $(x, y)$ and $(z)$, for the forward prediction problem. The prediction accuracy of the trained model is illustrated below. Figure 3g, h shows the density scatter plots of the ground-truth versus predicted values of $(x, y)$ and $(z)$ of 1 million sampling points randomly picked from the validation set (180,000 shapes, each with 256 sampling points). In both cases, the datapoints are mainly concentrated on the regression line, while those for $(z)$ are qualitatively more scattered than those for $(x, y)$. Quantitatively, the prediction achieves the $R^2 > 0.999$ for $(x, y)$ and $R^2 = 0.995$ for $(z)$, implying excellent accuracies for all three coordinates. Leveraging the symmetries can further enhance the prediction accuracy, achieving $R^2 \approx 0.999$ for $(z)$, as detailed in "Methods". Figure 3i further shows comparisons of the ground-truth (gray) and ML-predicted (colored) shapes for nine datapoints randomly picked from the validation set, where the contour visualizes the prediction error $\Delta r_i$ of each sampling point $i$, which is defined as the distance between the predicted and true positions ($i = 1, 2, \ldots, 256$). The maximum $\Delta r_i$ values for these shapes (marked in Fig. 3i) are significantly less than the edge

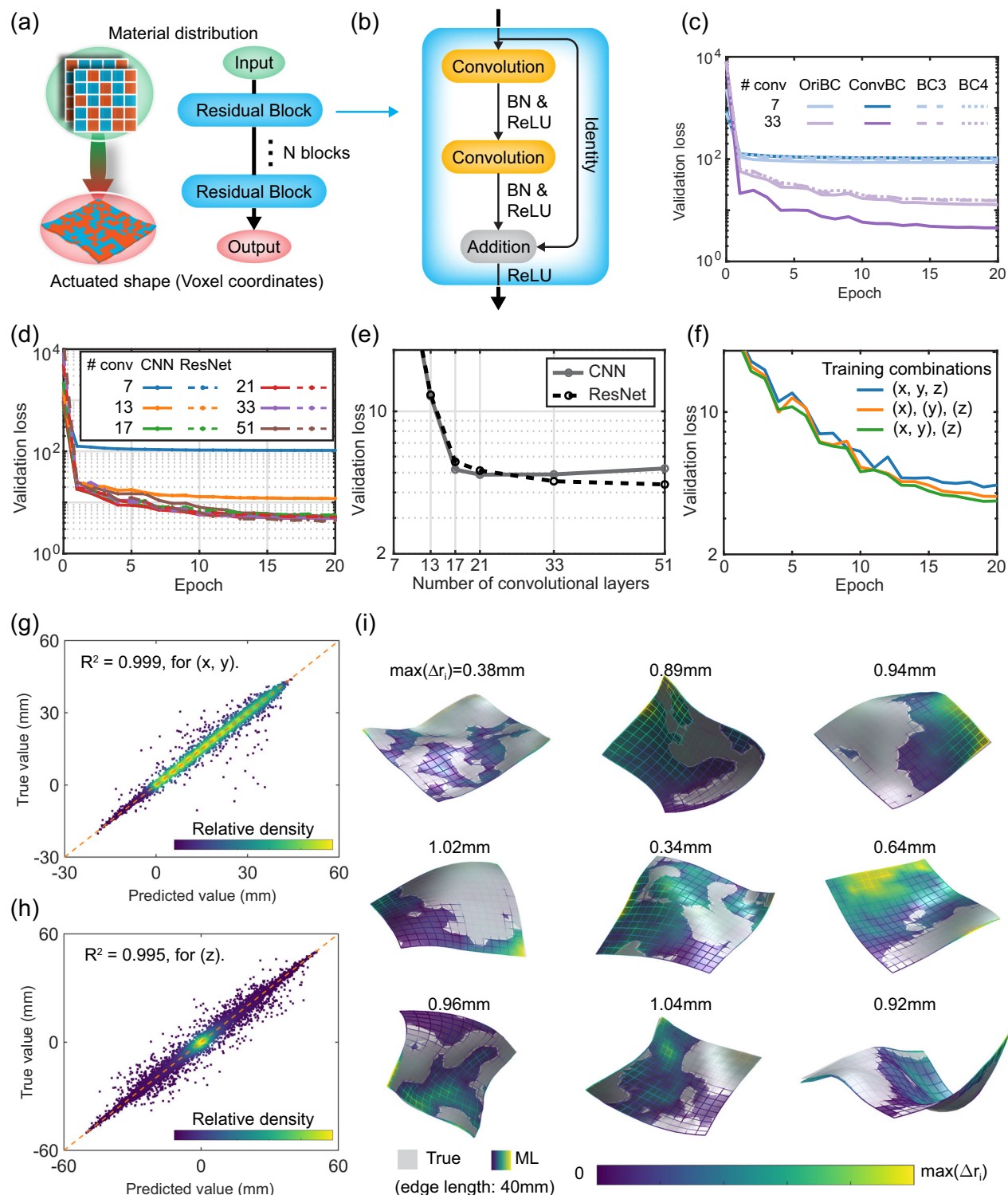

**Fig. 3 | Architecture and performance of the machine-learning (ML) model.**
**a** Architecture of the residual network (ResNet). **b** Architecture of the residual block. BN stands for batch normalization and ReLU the rectified linear unit. **c** Effects of four boundary conditions (BCs) on the ResNet performance. **d** Validation loss versus epochs showing the training progress of plain convolutional neural network (CNN) and ResNet for different numbers of convolutional layers. **e** Final validation loss for the cases in (**d**). **f** Effects of training groups on the ResNet performance with 51 convolutional layers. **g**, **h** Density scatter plots of the true versus ML-predicted values of the coordinate (**g**) $(x, y)$ and (**h**) $(z)$. The color indicates the relative point density. **i** Comparison of the ground-truth (gray) and ML-predicted (colored) shapes for datapoints randomly picked from the validation set. The color contour visualizes the prediction error $\Delta r_i$ of each deformed plate.

length (40 mm) of the plate, demonstrating an excellent agreement between the ground-truth and predicted shapes even for large deformations. Moreover, to evaluate the model's generalization ability, we build additional datasets that differ significantly in pattern types from the existing fully random and island datasets, as detailed in Supplementary Note 5. The distinction across datasets is illustrated through their example patterns (Supplementary Fig. 6) and statistics (Supplementary Fig. 7). Our ML model exhibits excellent performance on the

new datasets, and incorporating the new datasets into the training set slightly improves the performance (Supplementary Fig. 8). These demonstrate that our existing ML model has strong generalization ability.

In addition to the accuracy, the prediction speed of the ML (ResNet) model is also examined. We perform benchmark tests on the time cost of the ML and FE for 1000 shape predictions of randomly generated designs, either the fully random designs or the island designs. All tests are based on a single CPU core (Intel Core i9-10900) and one GPU (NVIDIA Quadro P620). As shown in Table 1, the FE prediction requires 11 and 28 h for the two designs, with the island designs being slower. In contrast, the ML model only takes 3.6 s regardless of the design type, which is much faster than the FE. These results demonstrate high efficiency of the ML model in the forward prediction problem. Also, as indicated in our previous work[40], high speed and high accuracy are critical for the inverse design problem.

## Inverse design approach based on ML: a global-subdomain strategy

Next, we introduce the ML-based approach for the inverse design of material distribution based on target actuated shapes. As discussed above, the design space for 2D active plates is enormous. Even with the ML model achieving high prediction accuracy, it still cannot directly handle the fundamentally different inverse design problem where distinct designs could yield similar actuated shapes. To address this challenge, a global-subdomain design strategy is adopted, which first optimizes all voxels globally to obtain a provisional optimal solution and then adjusts voxels in a subdomain with relatively large local errors

to further refine the solution (Fig. 4a). In particular, the subdomain is identified to contain $N_{sub}$ pixels with the highest local errors, which may consist of one or multiple region(s) depending on the spatial error distribution. Each pixel corresponds to $N_z$ design voxels in the thickness direction. All design voxels within the entire subdomain will be optimized simultaneously. Note that the subdomain design step may consist of multiple sub-steps. In each sub-step, the subdomain is re-identified based on the shape errors of the optimal design of the previous sub-step. The sub-step is repeated until the achieved solution converges (or cannot be further improved).

We then propose two specific algorithms, the GD (gradient-based) and the EA (gradient-free) for the optimization. As shown in Fig. 4b, in general, an optimization procedure consists of evaluating the design(s) using the ML forward model and generating new candidate design(s) using either GD or EA algorithm until an acceptable solution is found or a critical number of generations $N_{gen}$ is reached. The two algorithms are thus termed ML-GD and ML-EA, which are detailed in "Methods".

It is noted that either ML-GD or ML-EA can be used in the global design step. In contrast, the subdomain design will adopt the EA only. This is because compared to the global design, the reduced design domain (and thus design space) facilitates the random search for better designs in an EA process. In a GD process, however, the gradients with respect to all voxels always need to be tracked and the gradient-based design update is already efficient, thus the subdomain design does not improve the optimization efficiency.

## Design results for FE-derived target shapes based on given voxel patterns

We first consider the FE-derived target shapes generated by intuitively or randomly designated voxel patterns. More specifically, we generate intuitive or random designs, and the corresponding actuated shapes obtained through FE simulations are used as the targets. Albeit still challenging, this is a relatively simpler case as the shape obtained from FE simulations can possess features, such as surface continuity and smoothness, upon which the ML model is trained. In addition, although our ML model uses $15 \times 15 \times 2$ voxels, we specify the intuitive or random design in a grid with coarser in-plane resolution such that each grid consists of multiple voxels ($\geq 2 \times 2$), but perform the inverse design at full voxel resolution. In this way, there is more room for the ML voxels to vary while achieving a shape that approximates the target. Figure 5 shows the inverse design results for three different FE-

**Table 1 | Time cost for 1000 shape predictions with ML (ResNet) and FE for different types of design: 's' for seconds and 'h' for hours**

| Time cost for 1000 predictions | | |
|---|---|---|
| Design type, $N_x \times N_y \times N_y$ | ML (ResNet) | FE |
| Fully random, $15 \times 15 \times 2$ | 3.6 s | 11 h |
| Island, $15 \times 15 \times 2$ | 3.6 s | 28 h |

For the purpose of dataset generation, the FE simulations can be run in parallel to significantly reduce the time cost.

The estimation for time cost is based on a single CPU core (Intel Core i9-10900) and one GPU (NVIDIA Quadro P620).

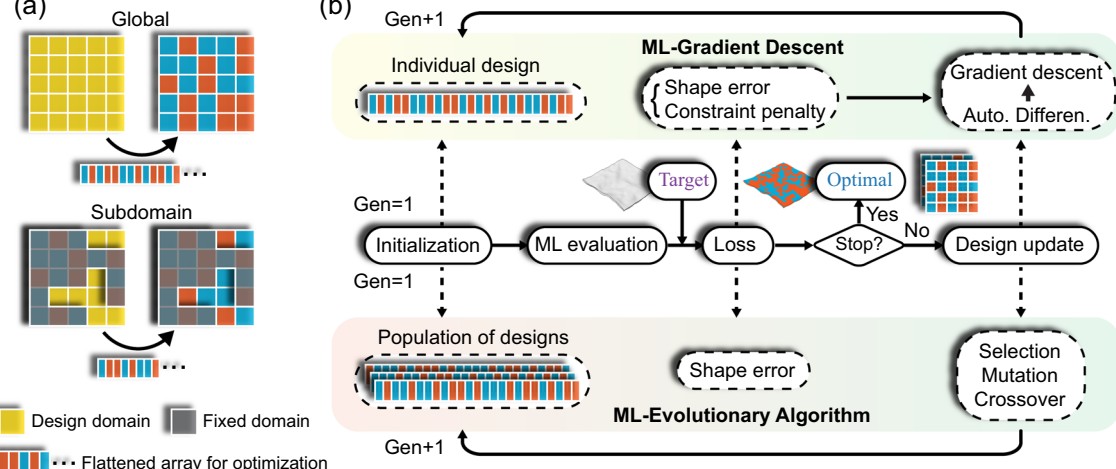

**Fig. 4 | Inverse design approach based on ML. a** Schematic of the global-subdomain design approach. **b** Schematics of ML-GD and ML-EA optimizations. The solid arrows are used for the main logical flow or direct connections, while the dashed arrows connect objects with explanatory relationships. Regardless of the algorithm, for consistency, each round of optimization is termed a "generation" (i.e., "Gen" in the figure) although "iteration" is more commonly used in GD.

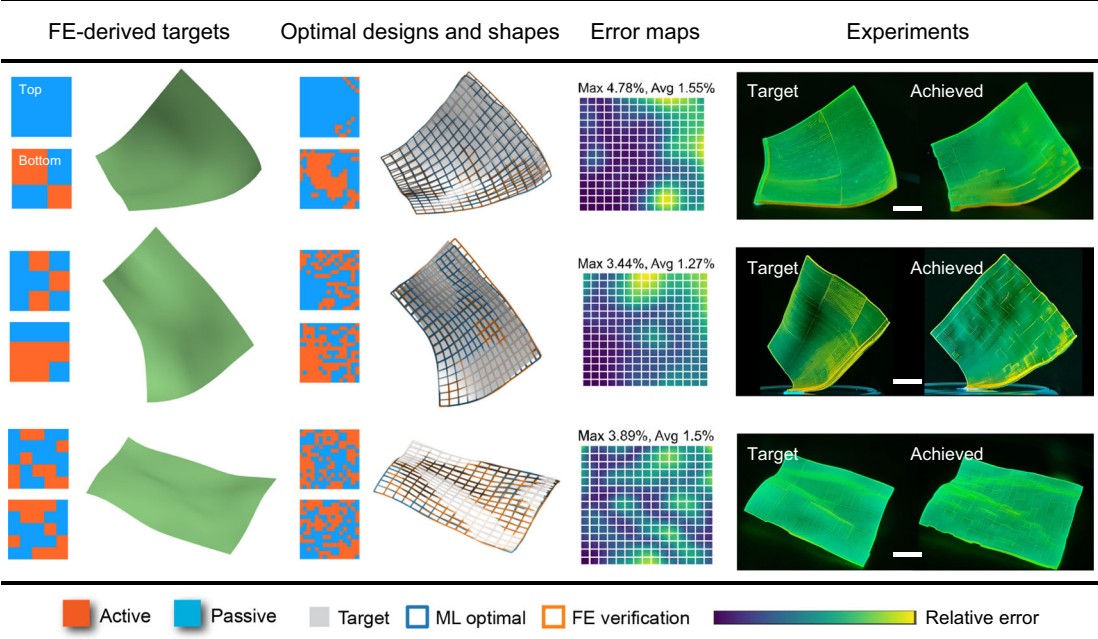

**Fig. 5 | Inverse design results for the FE-derived target shapes based on given voxel patterns.** From left to right: intuitive or random designs, the corresponding FE-derived target shapes (green), optimal designs, the corresponding actuated shapes predicted by ML (blue grid) and FE (orange grid) plotted against the target (gray surface), approximation error maps of the actuated shapes by FE with the maximum and average error values (relative to the edge length), and experimental comparisons of 4D-printed, actuated shapes of the given and optimal designs. Experimental results for the three targets are also shown in Supplementary Movies 1–3, respectively. All scale bars: 10 mm.

derived target shapes. As will be detailed later, we have performed different design trials for each target shape, and Fig. 5 only reports the best-achieved results. The given designs, target shapes, optimal designs, optimal shapes compared with targets, error maps, and experimentally 4D-printed, actuated shapes of both given and optimal designs, are shown in seven columns (from left to right). In column 4, since the optimization is entirely based on the ML model, the resulting optimal shape (i.e., the ML-predicted shape of the optimal design) is termed as the "ML-optimal shape" (blue grid). Also, since the ML always has prediction errors, we then use FE to evaluate the "true actuated shape" (orange grid) of the optimal design. To quantify the design accuracy, the approximation errors of the two optimal shapes against the target shape are obtained, using the absolute distance of each sampling-point pair in the two shapes (i.e., $\Delta r_{ij} = \sqrt{(x_{ij} - \hat{x}_{ij})^2 + (y_{ij} - \hat{y}_{ij})^2 + (z_{ij} - \hat{z}_{ij})^2}$, where $i, j = 0, 1, ..., 15$) divided by a reference length (taken as the edge length, 40 mm). In column 5, only the error map of the FE true actuated shape is shown for brevity. In columns 6 and 7, all printed square sheets are initially flat and only the actuated shapes are shown. Experimental details can be found in "Methods". As shown in Fig. 5, for all three targets, excellent agreement is achieved between the target (gray surface), the ML-optimal shape (blue grid) and FE true actuated shape (orange grid), which is quantitatively verified by the error map. The experimental actuated shapes of both given and optimal designs further validate such agreement (Fig. 5 and Supplementary Movies 1–3).

Before turning to other types of target shapes, we proceed to investigate the performance of our design approach in more detail. We first focus on the first target (row 1 of Fig. 5) but present complete design results for two separate global-subdomain trials (Supplementary Fig. 9), one with ML-GD for the global design and ML-EA for the subdomain design (Supplementary Fig. 9b), the other with ML-EA for both design steps (Supplementary Fig. 9c). In the global step, ML-GD and ML-EA give different optimal designs, but both result in the shapes that agree well with the target. Using the error map of the global step,

either with ML-GD or ML-EA, the subdomain is identified, and the subsequent optimization further improves the result. It is interesting to see that the subdomain located in the free corners is beneficial for the design improvement (Supplementary Fig. 9), which is also observed in the cases with other target shapes (Supplementary Fig. 10). This is again due to the spatially sequential dependency of the shape, which further justifies the use of distance-weighted loss function (Eq. (12)) in the global step, as the optimal solution is more likely to have smaller errors near the fixed boundary while larger errors near the free boundary.

It is also seen that in the global step, the optimal design by ML-GD is slightly better than that of ML-EA for this target (Supplementary Fig. 9). To check if this is correct in general, we next study the second target (row 2 of Fig. 5) and compare the performance of ML-GD and ML-EA (Supplementary Fig. 11), where the global step is the focus. Since ML-GD may be affected by the initial solution, to make fair comparisons, four ML-GD optimizations are performed using different initial solutions, i.e., all-passive ("0"s), all-active ("1"s), all-neutral ("0.5"s, unphysical values), and random designs (Supplementary Fig. 11). The results show that the ML-optimal shapes of all five cases agree remarkably well with the target, exhibiting a similar magnitude of errors (column 4 of Supplementary Fig. 11). This demonstrates the similar accuracy of two optimization algorithms. However, due to inherently random ML prediction errors, the approximation errors of the FE true actuated shape against the target are slightly more different across the five cases (column 5 of Supplementary Fig. 11): the ML-EA achieves the best design, by which the FE true actuated shape even outperforms the ML-optimal shape slightly; the ML-GD with the all-active initial design performs worst. Despite the error due to ML predictions, the five design trials all achieve satisfactory design results, demonstrating high accuracy of the ML model. Based on the results of the two targets (Supplementary Figs. 9 and 11), the performance of ML-EA and ML-GD is case-dependent. Furthermore, it is also seen that with ML-GD, the optimal design is highly sensitive to the initial solution, which is expected for such a highly nonlinear inverse problem. In

**Table 2 | Time cost for the inverse design: 'min' for minutes and 'h' for hours, 'gens' for generations**

| Algorithm or design | Time cost |
|---|---|
| ML-GD (1000 gens) | 3 min |
| ML-EA, global (100 gens) | 12 min |
| ML-EA, subdomain (5 gens) | 0.6 min |
| FE-EA (100 gens) | 2200–5600 h (estimated) |
| FE-GD (1000 gens) | 11–28 h (estimated) |
| Row 1 of Fig. 5 | 3.6 min |
| Row 2 of Fig. 5 | 12 min |
| Row 3 of Fig. 5 | 13.2 min |

The estimation for time cost is based on a single CPU core (Intel Core i9-10900) and one GPU (NVIDIA Quadro P620).

particular, the solution tends to vary away less from the initial one to achieve an optimal shape approximation, implying that it easier to identify a local minimal closer to the initial point in the high-dimensional loss-design parameter space.

Note that the optimization iterations directly use the ML predictions whose error will be inevitably forwarded to the design process. Nonetheless, excellent agreement is achieved between the optimal and the target shapes, demonstrating the high performance of the ML-based inverse design approach. In addition, the ML-GD achieves the optimal design in ~3 min through 1000 generations, and the ML-EA in ~12 min through 100 generations, the latter requiring roughly 200,000 evaluations of different designs. It can be inferred that for the same target shapes, the FE-EA[22] would consume more than 2200–5600 h (91–233 days), which are estimated using the benchmark time cost for FE based on a single core (see Table 1). Using the same benchmark, the FE-based GD (FE-GD) would require 11–28 h, not considering the time for gradient computation. The time cost for the inverse design with different algorithms are summarized in Table 2. The time for identifying a subdomain is less than 0.1 s and thus not shown. As the results shown in Fig. 5 are the best-achieved ones from different design trials, the time cost for different targets can be different, which are also given in Table 2. Despite the difference in time cost, the performance of ML-EA and ML-GD is case-dependent. The proper selection of the algorithm is in general nontrivial and dependent on the specific target shape. Therefore, in the later examples, we use the global ML-GD and the global ML-EA, both followed by subdomain ML-EA, and report the final optimization results.

## Design results for algorithmically generated target shapes

We now consider the algorithmically generated target shapes, which are more challenging for the optimization. This is because the ML-predicted shapes are represented by $16 \times 16$ grid points, and hence it is difficult to well define a target shape, i.e., to accurately give the target grid points that is physically attainable (i.e., that the actuated shape can achieve). For example, problematic sampling of grid points (too large or too small grid spacing) may cause difficulties in the optimization. Therefore, to make the problem tractable, we generate the target surface $(\hat{x}_{ij}, \hat{y}_{ij}, \hat{z}_{ij})$ based on a uniform, reference grid $(\hat{u}_{ij}, \hat{v}_{ij})$ on a square region (see details in "Methods"). Similar to our previous work[40], we then scale the generated surface such that its mean edge length is 41 mm (original length 40 mm with half of the thermal strain 0.05). This approximation may still give errors in the target such as those in grid point spacing, but the resulting surface can serve as a reasonable target for the optimization.

Figure 6 shows optimization results for multiple generated target surfaces with specific forms given in Table 3 (see "Methods" for descriptions). The target surfaces, optimal designs, optimal shapes compared against targets, error maps, and experimentally 4D-printed,

actuated shapes are shown in five columns (from left to right). The color in column 1 indicates the height. In column 5, all printed square sheets are initially flat and only the actuated shapes are shown. The first two targets are bow shapes that share the same parabolic feature while having different heights (rows 1 and 2). Such parabolic shapes are not easy to achieve through intuitive designs but are well captured by the ML-optimal designs, as seen in columns 2 to 4. In addition, the two shapes with distinct heights are not achieved through different magnitudes of actuation strains in active voxels, but rather through distinct pattern designs on the identical $15 \times 15 \times 2$ space, which also implies a nontrivial task. The experimental results eventually validate our optimal designs (column 5). Next, we further consider two nonuniform bending target shapes, also with identical form but varying heights (rows 3 and 4). Optimal designs are again obtained for the two distinct targets based on the same magnitude of actuation strain in active voxels and are further validated by experiments. Note that these target shapes (rows 1 to 4) are all developable surfaces, which cannot be achieved by the design approaches based on metric changes[47] but can be achieved by the non-intuitive voxel-level design with heterogeneities in the thickness direction. In the latter case, our approach proves to be very useful.

In the last example, we consider a non-developable target surface, the twisted parabolic shape (row 5 of Fig. 6)[18]. Our approach again yields the optimal design and the corresponding actuated shapes predicted by ML (blue grid) and FE (orange grid) agree well with the target (gray surface). Experiments are then conducted to validate the optimal design, where, to facilitate the validation, we compare the 4D-printed, actuated shapes with the 3D-printed target, which achieve remarkable agreement (Fig. 6 and Supplementary Movie 4). For the same case, we further implement our design in a smaller AC plate halved in size. As shown in Supplementary Movie 5, the printed sheet morphs upon heating and eventually achieves the target, which validates our design on such a smaller length scale. Moreover, we print our optimal design using two additional material systems that employ distinct actuation mechanisms and successfully achieved the target shape change in both cases (see Supplementary Movie 6 for the shape-morphing process and the actuated sheet). Details on the length scale and material systems are provided in "Methods".

These results demonstrate the great accuracy and efficiency of our inverse design approach in achieving algorithmically generated, developable and non-developable surfaces that may not have corresponding true designs and where the generated target points may involve problematic geometric features (e.g., spacing in grid) due to sampling. Moreover, our design approach demonstrates general applicability across various material systems, actuation mechanisms, and length scales.

## Design results for irregular target shapes

Next, we consider the irregular target shapes. In this case, the challenge in well defining a target becomes particularly severe. First, as discussed above, it is hard to appropriately specify the grid points with physically attainable spacing. Second, it is even harder to give the boundary of the target surface. In extreme cases, general irregular surfaces may involve boundaries that are physically unattainable by a square sheet, which would make the optimization intractable. To resolve these difficulties, we use a patch representation rather than the grid point representation for the target surface, and this new representation allows for extracting the surface normal of each patch and thus using a new measure of approximation errors (or loss) based on the normal distance of the achieved grid points to the target surface (Fig. 7a). This is schematically illustrated in Fig. 7a, where the black lines denote the measure of approximation errors, or distances between the target (gray surface, represented by purple points (top) or patches (bottom)) and the achieved surface (represented by blue

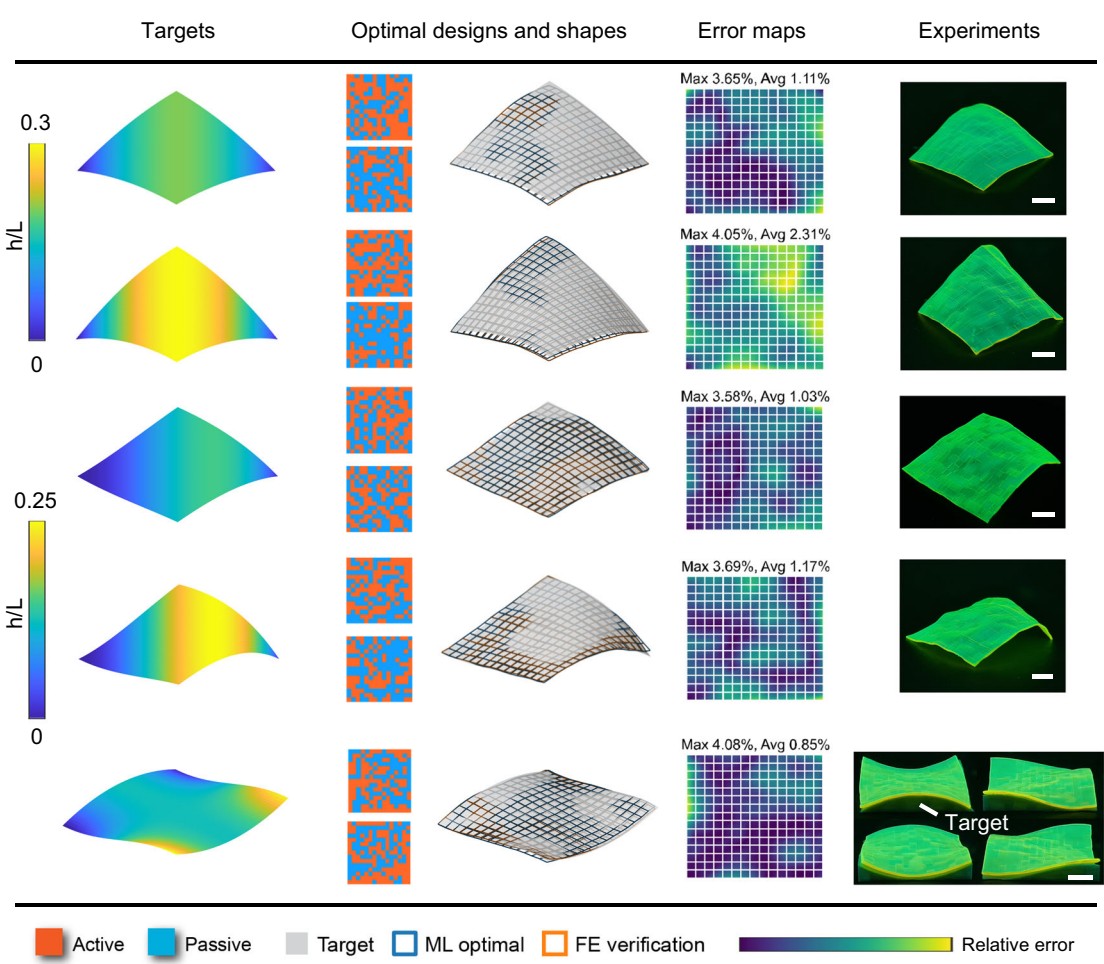

**Fig. 6 | Inverse design results for the algorithmically generated target shapes.** From left to right: target surfaces, optimal designs, the corresponding actuated shapes predicted by ML (blue grid) and FE (orange grid) plotted against the target (gray surface), approximation error maps of the actuated shapes by FE with the maximum and average error values (relative to the edge length), and experimentally 4D-printed, actuated shapes of the optimal designs. In experiments of row 5, the target shape is 3D-printed to facilitate the comparison. All scale bars: 10 mm.

points). Therefore, the new loss can be expressed as

$$\mathcal{L} = \frac{1}{(N_x+1)(N_y+1)} \sum_{i=0}^{N_x} \sum_{j=0}^{N_y} w_{ij} \left| d_{ij}(\hat{\mathbf{S}}) \right|^2 \qquad (6)$$

where $d_{ij}(\mathbf{S})$ denotes the distance of achieved point $(x_{ij}, y_{ij}, z_{ij})$ to the target surface $\hat{\mathbf{S}}$. With the new target representation and loss function, there are no strict requirements for the appropriate boundary of targets or sampling of grid points, as the optimization is essentially to achieve an actuated surface or patch that conforms to the target surface.

**Table 3 | Mathematical forms for algorithmically generated targets: $x = u$, $y = v$, and $z(u,v)$**

| Design | | Form of $z(u,v)$ |
|---|---|---|
| Fig. 6 | Row 1 | $z = 0.2(u+v)(2-u-v)$ |
| | Row 2 | $z = 0.3(u+v)(2-u-v)$ |
| | Row 3 | $z = 0.1(u+v)^2(2-u-v)$ |
| | Row 4 | $z = 0.2(u+v)^2(2-u-v)$ |
| | Row 5 | $z = -0.1(2u-1)^2 \sin[(v-0.5)\pi] + 0.1$ |
| Fig. 7b | | $z = 0.2(u+v)(2-u-v)(-\frac{4}{19}u+\frac{15}{59}v)$ $\cdot(-5u^2-\frac{3}{4}u+1)(\frac{10}{3}v^2-\frac{2}{5}v+1)$ |

Figure 7b shows an example to illustrate the new strategy. We define the target again based on the reference grid $(\hat{u}_{ij}, \hat{v}_{ij})$ on a square region, but a rather irregular shape is specified using the form given in Table 3. Due to the irregularity, the edge length scaling is inappropriate as the lengths of four edges are unequal; also, the grid points sampled based on the reference grid have the problematic spacing. As a result, using the grid point-based target shape and loss function, the optimal shape is missing where the target has complicated features. In contrast, using the triangular patch-based target shape, the optimal shape well captures the target. In this case, we do not expect the achieved shape to approximate all grid points, e.g., the corner points, but rather to conform to the target surface.

To further demonstrate the new strategy, we show two examples of more irregular target shapes. In the first example, we crumple and unfold a square piece of paper, which naturally forms a random shape and is then scanned to serve as the target (Fig. 7c–i). The fine unsmooth features (or kinks) make it difficult to well define a grid point-based target shape, i.e., to determine the target points in which the actuated (optimal) shape should be located. As shown in Fig. 7c–ii–iv, here, using the triangular patch-based target and the loss function Eq. (6), the optimization achieves excellent agreement among the target (gray surface), the ML-optimal shape (blue grid) and the FE-predicted shape (orange grid). Further, experimental results show remarkable agreement between the experimentally actuated shape

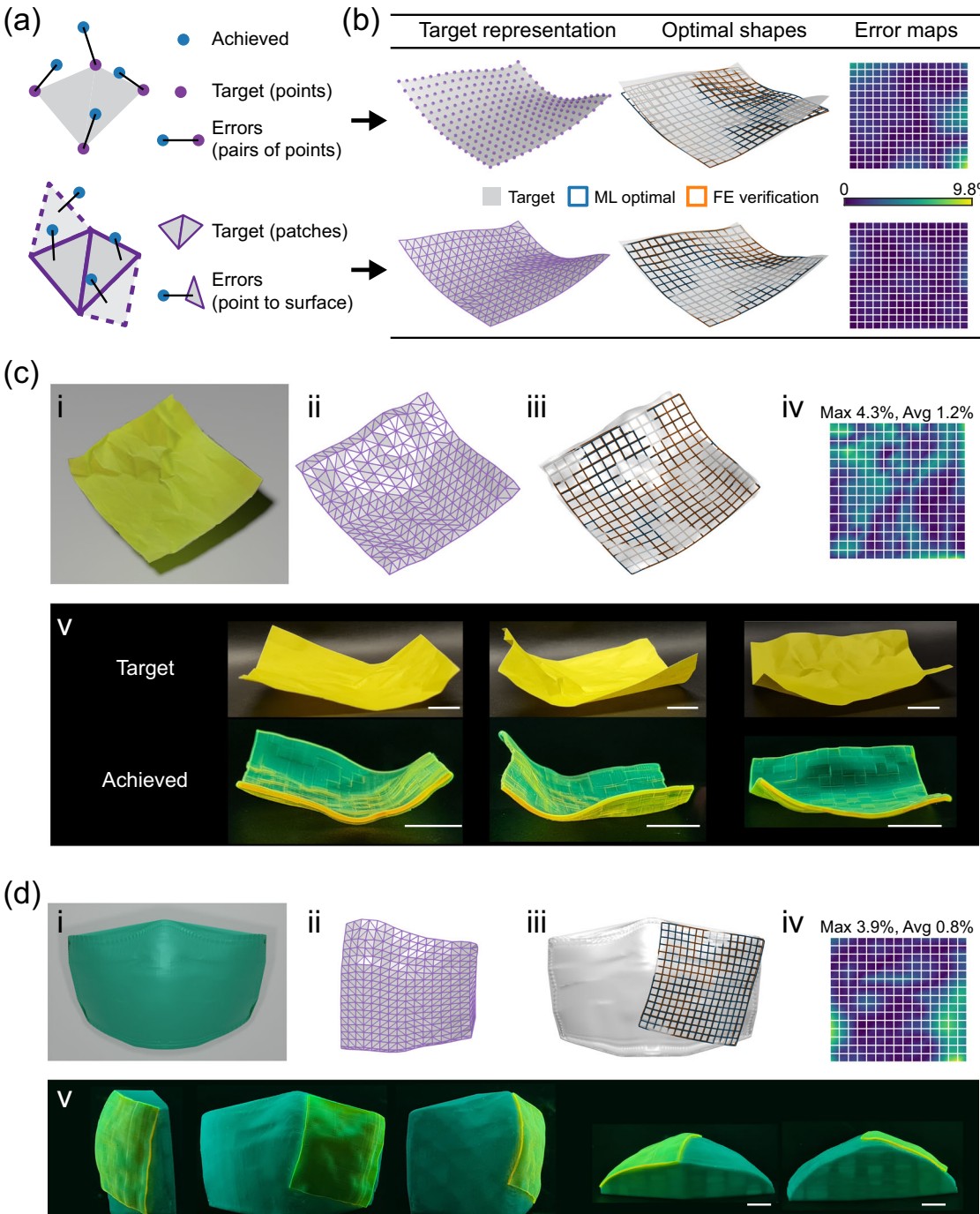

**Fig. 7 | Inverse design results for irregular target shapes or patches. a** Two representations of the target surface, the grid point representation (top) and patch representation (bottom), and corresponding approximation errors for the loss calculation, the point-pair distances (top) and the normal point-to-surface distances (bottom). **b** Comparison of design results obtained through the two strategies for an algorithmically generated irregular target shape. **c** Design result for the scanned target of a crumpled paper. A complete design-fabrication process is shown in Supplementary Movie 7. **d** Design result for the patch target of a surgical mask. Show in (**c**, **d**) include the raw target (i) and patch representation (ii), the optimized shape (iii) and error map (iv), and the 4D-printed, actuated shape (v). All scale bars: 10 mm. All the error maps are measured for the FE-evaluated actuated shapes versus the target, and the error values are relative to the edge length.

and the original crumpled paper, validating the ML-optimal design, as seen in Fig. 7c–v and Supplementary Movie 7. This movie also shows a complete design-fabrication process from the paper crumpling to final 4D-printed shape. Note that for targets with severe unsmoothness, one can pre-smooth the target before the design, as shown in Supplementary Fig. 13. Here, without pre-smoothing, our algorithm implicitly smooths the target during the design. In the second example, we scan a surgical mask as our target shape, which has irregular boundaries

(Fig. 7d–i). To reduce computational cost, we take a part (more than half) of the mask surface as the target for the actual optimization (Fig. 7d–ii). Figure 7d–iii–iv show the optimization results where the actuated shapes predicted by ML (blue grid) and FE (orange grid) excellently conform to the target (gray surface). Experimental results further validate that the final actuated shape indeed conforms to the 3D-print target (surgical mask), as shown in Fig. 7d–v and Supplementary Movie 8. These results demonstrate the capability of our

design approach in achieving a variety of irregular target shapes, through the direct approximation or the conforming patches. Equipped with our approach, the voxel-level design strategy is also demonstrated to be promising in facilitating applications of 4D-printed shape-morphing structures, such as the customized smart mask.

## Discussions and potential improvements
Our approach that integrates ML with certain optimization algorithms (GD or EA) demonstrates high efficiency in solving inverse design problems in a complex design space. Although training the ML model requires a large number of FE simulations (~56,000), this number is much lower than that of the shape predictions (200,000) needed in an EA for complicated target shapes, and therefore ML-EA is always more efficient than FE-EA. In addition, ML allows for rapid shape predictions and efficient gradient computations via AD, thereby enabling the computationally low-cost GD process. This thus offers new possibilities for addressing the challenges faced by TO (which can be seen as FE-GD), the local minima problem and the complicated gradient derivation[32]. More importantly, once an accurate ML model is trained, it can be reused for efficient inverse designs of many different target shapes. Therefore, the time cost for obtaining the ML model is offset by the significant time savings of inverse design, when compared to conventional design methods[18,22,23]. Furthermore, both data generation and model training can be further parallelized to improve computational efficiency. For example, in our 10-group parallelized FE simulations, the use of 10 cores of a CPU does not represent a substantial computational resource. By using more CPU cores (e.g., 60), which are readily available across multiple computers or within a cluster, the FE simulation time can be significantly reduced. This also suggests the feasibility of scaling our design space (e.g., to $45 \times 45 \times 2$ or $90 \times 90 \times 2$ voxels) without leading to prohibitive time costs.

There are potential improvements to our approach. First, we demonstrate the possibility to design irregular target shapes that are attainable by a square sheet. For the cases where the initial shape is not square, e.g., triangular or rectangular, or those with finer features that cannot be adequately captured by the current design space ($15 \times 15 \times 2$ voxels), our design approach may still be applicable without training a new ML model. For example, one can use cutting if the initial shape is triangular or rectangular, or combine multiple conforming patches to construct surfaces containing a larger number of voxels. These potential modifications could expand the design space of our current ML model and are our ongoing efforts. This idea was demonstrated for AC beams in our recent study[41]. Second, our printed AC sheets exhibit some unsmooth features at the voxel boundaries, which arise from variations in the curing distance for different grayscale levels of DLP printing. This issue could potentially be addressed through the optimization of printing parameters[66] to achieve more accurate printing of our optimized material distributions. This is our ongoing work. Third, our ML model is purely supervised by data. A physics-informed ML model[67,68] that incorporates appropriate physical constraints into the loss function could potentially reduce the amount of data needed, which will be explored in our future research.

It is worth noting that the proper selection of boundary conditions (BCs) is important in the forward and inverse problems of active plates. The original BCs preserve all physical symmetries and thus enable easy data augmentation. On the other hand, the converted BCs endow the deformed shape (in terms of coordinates) with the spatially sequential dependency, which is proven to be beneficial for the ML prediction and the ML-EA optimization. These insights are useful for the application of ML model to other shape-morphing problems of AC structures.

In addition, our design approach is generally applicable across various material systems, actuation mechanisms, and length scales. This also implies the general applicability to other voxelated printing techniques. The ML model and design approach can also be extended to the cases with multiple (>2) material phases which provides greater design flexibility. Therefore, our approach will be useful for motivating the design for various 4D-printed ACs.

## Discussion
We present an approach for shape-change prediction and inverse design of a 4D-printed active composite plate based on ML and EA. A residual net (ResNet) based ML model is utilized to predict the shape change based on the material distribution of the active plate. By combining key ingredients such as the data augmentation by symmetries and the proper selection of boundary conditions, the ML model achieves excellent accuracy on the validation set with $R^2$ reaching 0.999 for the coordinate ($x$, $y$) and 0.995 for ($z$). In addition, the ML prediction is much more rapid and computationally inexpensive than the FE model. We then incorporate the trained ML (ResNet) model into both the GD (gradient-based) and the EA (gradient-free) algorithms for the inverse design of the material distribution based on the desired shape change. The ML accelerates the GD by enabling not only the fast-forward prediction but also the efficient computation of exact gradients via AD. On the other hand, the ML allows the EA to search solutions in a large design space that is impossible to be explored by FE simulations. As a result, using a global-subdomain design strategy with the two algorithms, optimal designs in terms of material distributions for a variety of numerically generated target shapes (or surfaces) are rapidly achieved. By further combining ML-EA with a patch representation of the target and a normal distance-based loss, optimal designs are achieved for multiple irregular target shapes and validated by computations and experiments. Therefore, our voxel-level inverse design approach empowered by ML paves the way for the intelligent design and fabrication of 4D printing, and is thus promising in facilitating broader applications of 4D-printed shape-morphing AC structures.

## Methods
### Finite element model
We perform FE simulations using the commercial software Abaqus (version 2018, Simulia, Providence, RI) to generate the dataset to be fed into the ML model. In the FE model, both active and passive materials are modeled using the incompressible neo-Hookean model with the same modulus. Expansion is achieved through the thermal expansion in the simulation, although it is not limited to thermal expansion. To achieve the expansion mismatch, the coefficient of thermal expansion is set to be 0.001 for active material and 0 for passive material. A 50 °C temperature increase is applied to the entire plate, upon which the active material phase is subjected to a linear strain of 0.05. The plate has a dimension of 40 mm × 40 mm × 1 mm, which is meshed into $45 \times 45 \times 4 = 8100$ C3D8H (eight-node linear brick, hybrid) elements. Mesh convergence study has been performed to ensure the mesh independence of FE results. The two materials are assigned to $N_x \times N_y \times N_z$ (i.e., $15 \times 15 \times 2$) voxels and each voxel has ($45/N_x$) × ($45/N_y$) × ($15/N_z$) elements. During the shape change, the potential self-contact of the plate is not accounted for. The building and running of the entire FE model are automated through a Python script, which reads in binary number arrays of '1's (active) and '0's (passive) and yields the corresponding actuated shapes.

### Dataset preparation and statistics
The FE model is used to obtain the true actuated shapes of 56,250 randomly generated designs, which include two types: 31,250 fully random designs and 25,000 island designs. Example patterns of each type are shown in Supplementary Fig. 1a. As these designs are represented by 3D binary arrays with a shape of $15 \times 15 \times 2$, the first type of design is simply obtained through the function *randint* of NumPy

library, giving random arrays without any pattern constraints; whilst the second type is obtained through certain combinations of random island images with connected domains of "1"s or "0"s, which are detailed in Supplementary Note 1. Next, FE simulations are performed using the original BCs (Fig. 2), and pairs of material distribution **M** and actuated shape **S** constitute the dataset, which is then augmented 16 times following the symmetric rules. Supplementary Fig. 2 illustrates the symmetries and augmented designs, whose actuated shapes can be calculated without the need for new simulations (see the caption). With the original BCs, the dataset is augmented 16 times, yielding 900,000 data in total, which is then converted to the other set of BCs. Finally, the dataset is split into training and validation datasets with fractions of 0.9 and 0.1.

For the dataset generation, we perform FE simulations in parallel across 10 groups, each using a single core of a CPU (Intel Core i9-10900), which takes about 95 h. This setup does not represent a substantial computational resource. In fact, our current CPU supports running 20 simulations simultaneously owing to its 10 physical cores with hyper-threading. Therefore, the FE simulation time may be significantly reduced depending on a user's computational resource. Moreover, this is faster than the benchmark tests of FE simulations shown in Table 1, which is based on a single CPU core.

We study the statistics of the dataset. For each data (pair of **M** and **S**), the maximum displacement is calculated. The distributions of the maximum displacement for different datasets are then obtained. First, we compare the distributions of the fully random dataset and the island dataset under both original BCs (Supplementary Fig. 3a) and converted BCs (Supplementary Fig. 3b). In both sets of BCs, the island dataset overall exhibits a larger maximum displacement than that of the fully random dataset. In converted BCs, for example, the mean value of the maximum displacements is 8.4 mm for the fully random dataset and 33.0 mm for the island dataset, respectively. This implies large displacements compared to the edge length (40 mm). Second, the training and validation datasets, which are based on the converted BCs, follow similar distributions (Supplementary Fig. 3c). This is expected since the two datasets (training and validation) are randomly sampled from the entire dataset.

## Construction and training of the ML models

Our ResNet consists of an image input layer, $N_{block}$ residual blocks, and a regression layer (Fig. 3a, b). For the input layer, a single material distribution has three dimensions ($x$, $y$, and $z$), which are modeled as two spatial dimensions and one channel dimension of an image, respectively. All convolutional layers are 2D convolutional layers with filter size of [3 3]. The implementation, training, and testing are conducted using MATLAB (2022a, MathWorks, Natick, MA). Before the training, all the input and output data are normalized using the z-score method, i.e., $x' = (x-mean(x))/std(x)$, where $x$ and $x'$ are the raw and normalized feature values, respectively, and mean is the mean value and std is the standard deviation. The randomly generated raw inputs (numerous "1" and "0") show a mean value of 0.5 and a standard deviation of 0.5. As a result, the input state "0" and "1" become "-1" and "1" after normalization. Such normalization is found to improve the network performance. Regarding the hyperparameters, the initial learning rate is set to 0.001, which decreases by multiplying a factor of $1/\sqrt{2}$ every 12 epochs. The training stops after the validation loss converges. The mini-batch size during training is set to 512. The adaptive moment estimation (Adam)[69] optimizer is used to train the network. The model training takes about 10 h on a single GPU (NVIDIA Tesla V100). The total time for preparing the dataset and training the ResNet model is thus about 105 h. In addition to ResNet, we also build the plain CNN and GCN models. Our CNN has the same architecture as ResNet but without skip connections. Details on our GCN implementation are provided in Supplementary Note 4.

## Leveraging symmetries to enhance the ML prediction accuracy

The symmetries can also be used to enhance the ML prediction accuracy. With a given design, three symmetric designs are obtained using the transpose ("Swap X,Y") and "Flip Z" operations as shown in Supplementary Fig. 2. The ML model is used to predict the shapes of the three transformed designs, which are then converted back into shapes corresponding to the original design. We thus have four predicted shapes for the same design, which are averaged to give the final prediction. Note that the other symmetries are not used, as the adopted converted BCs prevent us from recovering the shapes of the original design. Applying such an averaging approach to the coordinate ($z$) improves the prediction accuracy, increasing $R^2$ from 0.995 to 0.999. The prediction of ($x$, $y$) already achieves $R^2 > 0.999$, so the averaging approach is not used. Note that using symmetry to achieve higher accuracy will require four times the computational time of one regular prediction. Therefore, in our inverse design tasks, we use this enhanced ML model in the final evaluation of the achieved solution, but not in the optimization process.

## ML-GD approach for global design

In ML-GD approach (Fig. 4b), the pre-trained ML models (ResNet-51 for (x,y) and (z)) are used to predict the actuated shape. To minimize the error between the achieved shape and the target, we use the mean squared error (MSE), i.e., the $L_2$ norm of the shape difference divided by the number of coordinate data, as a loss term:

$$\mathcal{L}_1(\boldsymbol{\theta};\mathbf{M}) = \frac{1}{3(N_x+1)(N_y+1)}\sum_{i=0}^{N_x}\sum_{j=0}^{N_y}\left[(x_{ij}-\hat{x}_{ij})^2+(y_{ij}-\hat{y}_{ij})^2+(z_{ij}-\hat{z}_{ij})^2\right],$$

(7)

where ($x_{ij}, y_{ij}, z_{ij}$), components of (**x, y, z**) ( $= \mathcal{F}(\boldsymbol{\theta};\mathbf{M})$, see Eq. (3)), are the ML-predicted coordinates of all sampling points (voxel mesh points on the mid-surface) of an individual design **M**; ($\hat{x}_{ij}, \hat{y}_{ij}, \hat{z}_{ij}$) are the target coordinates of the corresponding points; $i$ and $j$ represent the indices for the grid points in the $x$- and $y$-directions; and $\boldsymbol{\theta}$ denotes the parameter set of the pre-trained forward model, $\mathcal{F}$. Here, the design variable $M_i$ (component of **M**) must be discrete values of -1 (passive) or 1 (active) to be physically meaningful (these are normalized values of 0 and 1, as described in "Construction and training of the ML models", "Methods"). This poses a discrete variable optimization problem, which is difficult as **M** has to be continuously updated in a GD process. In addition, unphysical values of **M**, in particular $|M_i| > 1$, may produce unexpected outputs $\mathcal{F}(\boldsymbol{\theta};\mathbf{M})$ in Eq. (7) during the optimization. To address these issues, we enforce the design **M** to be an output of the hyperbolic tangent function, such that $|M_i| < 1$,

$$\mathbf{M} = \tanh(\mathbf{M}')$$

(8)

where **M'** takes place of **M** to be the new design variable and can be continuously updated. Further, to enforce the discrete variable constraint, we propose a second loss term that penalizes the deviation of $M_i$ from its constrained values (−1 or 1),

$$\mathcal{L}_2(\mathbf{M}) = \frac{1}{N_{voxel}}\sum_{i=1}^{N_{voxel}}(1-M_i^2),$$

(9)

where $N_{voxel}$ is the number of design voxels. Adding up the two loss terms and using Eq. (8), the total loss function for the inverse problem is thus given as

$$\mathcal{L}_{GD}(\mathbf{M}') = \mathcal{L}_1 + \beta\mathcal{L}_2,$$

(10)

where $\beta$ is the weighting factor for the constraint penalty, which is a hyperparameter. We use $\beta = 0.1$ in our optimizations. Then, the goal of

the optimization is to minimize the loss function Eq. (10) by iteratively adjusting $\mathbf{M}'$, which can be expressed as

$$\min_{\mathbf{M}'} \mathcal{L}_{GD}. \tag{11}$$

The Adam optimizer is used in the ML-GD. The maximum number of generations (i.e., "iterations" commonly used in GD) is set as $N_{\text{gen}} = 1000$. Once the entire process is complete, the optimized $\mathbf{M}'$ will be processed through Eq. (8) and then rounded off to either −1 or 1. Note that a gradient-based approach requires computing the derivatives $\partial \mathcal{L}_{GD}/\partial \mathbf{M}'$, which could be complicated and time-consuming by the conventional approach. Here, the ML-based forward model involves differentiable operations only and thus naturally allows tracking of gradients via AD. Thus, exact $\partial \mathcal{L}_{GD}/\partial \mathbf{M}'$ values can be computed efficiently.

## ML-EA approach for global or subdomain design

EA is a population-based stochastic search technique that utilizes the principles of natural selection to seek optimal inputs producing desired outputs[70]. The ML-EA therefore performs the optimization on a population of designs or individuals (Fig. 4b). In a design generation, we use the pre-trained ML model to predict the actuated shapes and calculate the loss values of all individuals. The population is then evolved via the selection, mutation, and crossover operations, such that good individuals survive and reproduce whilst bad individuals are eliminated. More details can be found in Supplementary Note 7. In ML-EA, the loss function is modified from Eq. (7) to the following form

$$\mathcal{L}_{EA} = \frac{1}{3(N_x+1)(N_y+1)} \sum_{i=0}^{N_x} \sum_{j=0}^{N_y} w_{ij} \left[ (x_{ij}-\hat{x}_{ij})^2 + (y_{ij}-\hat{y}_{ij})^2 + (z_{ij}-\hat{z}_{ij})^2 \right], \tag{12}$$

which is still a measure of how close the actuated shape is to the target, but a weighting factor $w_{ij}$ for sampling points is introduced. We use $w_{ij} = 1/(i+j+1)$ for the global optimization step, where $i+j+1$ represents a topological distance of the sampling point from the starting point (i.e., the fixed point). Similar to ref. 40, this yields a distance-weighted loss function that favors to sequentially optimize the plate shape from the fixed boundary to the distant free boundary, such that the later improvement in the distant part would not affect the shape in the nearer part, thereby benefiting the EA iterations. In the subdomain optimization step, we use $w_{ij} = 1$ such that Eq. (12) recovers Eq. (7), the normal loss based on shape MSE. Note that no additional loss is included since the binary constraint of a design (array of '1's and '0's) can be naturally met under selection, mutation and crossover operations in an integer-valued EA process. To summarize, the goal of the optimization can be expressed as

$$\min_{\mathbf{M}} \mathcal{L}_{EA} \text{ subject to } M_i \in \{0,1\}. \tag{13}$$

The following EA parameters are used: population size (number of individuals) = 2000; $N_{\text{gen}} = 100$ for the global optimization and $N_{\text{gen}} = 5$ for the subdomain optimization.

## Materials and 4D printing

The ML-optimal designs are validated by the 4D printing experiments with an ink that enables volatilization for shape morphing[71]. The resin was formulated with a volatile monoacrylate component with a low boiling point (isobornyl acrylate) and a nonvolatile oligomer diacrylate (aliphatic urethane diacrylate) as a crosslinker. The grayscale digital light processing (g-DLP) printing technique is used to print the designed structure, where the degree of conversion (DoC) of the resin can be spatially controlled by the assigned grayscale level (and thus

light intensity)[15,40]. Our ML-optimal designs are converted into grayscale printing slices such that the active ("1") and passive ("0") voxels are printed using brighter (0% grayscale) and dimmer (60% grayscale) lights and thus lead to well-cured (higher-DoC) and partially cured (lower-DoC) material phases, respectively. The printed structure is then heated to facilitate the monomer volatilization. The partially cured phase ("0") contains more residual monomers that can volatize at elevated temperatures and thus shows more volume shrinkage than the well-cured phase ("1"). The mismatch of the shrinkage strain thus induces the shape transformation.

The photocurable resin is prepared by mixing isobornyl acrylate (IOBA, Sigma-Aldrich) and aliphatic urethane diacrylate (AUD, Ebecryl 8402, Allnex, GA, USA) in a weight ratio of 1:1. Then, 1 wt% photoinitiator (Irgacure 819, Sigma-Aldrich), 0.08 wt% photoabsorber (Sudan I, Sigma-Aldrich), and 0.04 wt% fluorescent dye (Solvent green 5, Orichem International Ltd., Hangzhou, Zhejiang, China) were added. The resin is thoroughly mixed before printing. After printing, the sample is placed in an 80 °C oven for 8 h to facilitate the volatilization of IOBA, inducing shape-shifting. Subsequently, the sample with the obtained shape is post-cured for 1 min on each side using a UV lamp to further cure the nonvolatile AUD residuals and stabilize the structure. This resin is used throughout the paper unless otherwise specified. Note that the material properties in experiments are different from those used in FE simulations. The printed two material phases show a modulus ratio of 0.053 and a strain mismatch of 0.057, while the FE (or ML) model assumes the identical modulus and a strain mismatch of 0.05. Such a property difference would result in different shape changes between ML predictions and 4D-printed parts. This issue can be resolved by retraining a new ML model based on practical material parameters and rerunning the design. Here, instead of retraining a new model, we adopt a strategy similar to that of our previous work[40,41], i.e., tune the print dimension to approximately compensate for the effect of property differences through an analytical model for the local curvature. Details are provided in Supplementary Note 9. This dimension modification strategy offers an efficient way to applying our ML model across different materials and length scales.

Alternatively, two additional grayscale DLP printable material systems are presented in Supplementary Movie 6. The first one, which undergoes deformation under the same heating conditions, consists of trimethylolpropane triacrylate (Sigma-Aldrich), Ebecryl 8402, and n-butyl acrylate (Sigma-Aldrich) in a weight ratio of 1:2:2, with the same loading of additives (1 wt% photoinitiator, 0.08 wt% photoabsorber, and 0.04 wt% fluorescent dye, the same for the second one). The second material system is composed of poly(ethylene glycol) diacrylate (Sigma-Aldrich) and 2-hydroxyethyl acrylate (Sigma-Aldrich) in a weight ratio of 1:1. Activation is achieved by swelling in acetone for ~7 min, followed by drying in air.

## Mathematical forms for algorithmically generated targets

The surface equations $x$, $y$, and $z$ are constructed from 2D parametric variables $u$ and $v$ that range between 0 and 1. For simplicity, all the complexity of the surface is confined to $z$, as such, $x(u,v) = u$, $y(u,v) = v$. We then use a group of simple functions to construct different target surfaces that satisfy certain boundary conditions (see Supplementary Note 10). The mathematical forms of generated targets are provided in Table 3. Using these forms, the target surfaces in terms of points, $(\hat{x}_{ij}, \hat{y}_{ij}, \hat{z}_{ij})$, are sampled based on a uniform, reference $16 \times 16$ grid, $(\hat{u}_{ij}, \hat{v}_{ij})$. Therefore, the target $(\hat{x}_{ij}, \hat{y}_{ij}, \hat{z}_{ij})$ may have nonuniform spacing depending on the specific forms, but can still serve as a reasonable target for the optimization.

## Reporting summary

Further information on research design is available in the Nature Portfolio Reporting Summary linked to this article.

## Data availability

The authors declare that the data supporting the findings of this study are available within the paper and its supplementary information files. The generated dataset and the pre-trained model are available in Kaggle[72]: https://www.kaggle.com/datasets/sunxiaohao/dataset-for-active-shapes-of-ac-plates. Source data are provided with this paper.

## Code availability

The codes for the dataset generation, the machine-learning model, and the inverse design are available in Zenodo[73]: https://github.com/XiaohaoSun/ML_4DP_AC_plates.

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

## Acknowledgements

H.J.Q. acknowledges the support of an AFOSR grant (FA9550-20-1-0306; Dr. B.-L. "Les" Lee, Program Manager) and a gift fund from HP, Inc. This research was supported in part through research cyberinfrastructure resources and services provided by the Partnership for an Advanced Computing Environment (PACE) at the Georgia Institute of Technology, Atlanta, Georgia, USA.

## Author contributions

X.S. and H.J.Q. conceived the concept. X.S. and H.J.Q. designed the computations. X.S. and C.F. performed the computations. L.Yue designed the experiments. L.Yue., L.Yu and C.A. performed the experiments. L.Yu performed the property characterization. X.S. wrote the initial draft. L.Yue and X.S. prepared the movies. L.Yu, K.Z., F.D. and R.R.Z. commented on the paper. X.S. and H.J.Q. edited the paper. H.J.Q. supervised the project.

## Competing interests
The authors declare no competing interests.
