## [Peer Review File · Nature Communications]

Machine learning-enabled forward prediction and inverse design of 4D-printed active platesREVIEWER COMMENTS

Reviewer #1 (Remarks to the Author):

This paper presents the design optimization of pixelated active composite materials, to achieve target deformation behaviors. To reduce the computational cost, the author first trained a ResNet model which performs fast prediction. Materials are then optimized using gradient descent and evolutionary algorithms based on neural network queries. Overall, this research is well presented with results validated by both numerical simulations and real experiments through 3D-printing. Implementation details are provided to allow reproduction. However, the contribution of the manuscript is more on the application of ML-based optimization on deformation optimization of smart composites, rather than the optimization algorithm itself, as similar methods have been seen in many literatures: CT Chen, GX Gu - *Advanced Science*, 2020, and S Lee, Z Zhang, GX Gu - *Materials Horizons*, 2022, etc. So it would be better if the author could elaborate more (in Intro) on the application side for motivation. There are also a couple of detailed questions regarding the modeling and optimization method used in this work, as below.

- The author trained a ResNet as a surrogate model to make fast inferences for different AC designs. The training process seems to be fully supervised. As the model is predicting the displacement (coordinates) of the voxels, is it possible to add in some physical constraint like continuity or smoothness of the deformed composite, which potentially reduce the amount of data needed?
- The AC in this work has a design space of size $15 \times 15 \times 2$. However, for most topology optimization problems in literature, the design space is significantly larger, some are like 100×100 or even larger. Is it still viable to train a surrogate model for design optimization when each simulation takes hours to accomplish? It would be great if the author can add in some discussion regarding the scalability of surrogate model based optimization.
- From the optimization results in Fig. 5 and 6, the activated shapes indeed match well with the target shape macroscopically. But the deformed materials are not as smooth as the target shape microscopically. Is this problematic in application? How can this issue be potentially addressed?
- Surrogate model based optimization can be problematic as the optimization process not only maximize/minimize the design objective, but may also converge at someplace with maximized prediction errors. In this study, this phenomenon isn't obvious as the model is trained on a rich dataset and have an extremely high R-squared value. But how would the optimization strategy be improved if the surrogate model gives certain bias from ground truth, say only 85-90% R-squared value?

Reviewer #2 (Remarks to the Author):

This paper presents a machine learning and evolutionary algorithm enabled approach for designing and predicting shape changes of 4D printed active composite plates. A residual network based machine learning model is developed to predict shape changes with great accuracy and efficiency. The model is then combined with gradient descent and evolutionary algorithms to achieve inverse design of optimal material distributions for desired shape changes, which is significantly accelerated by machine learning. A global-subdomain design strategy is proposed to obtain optimal designs by first globally optimizing all voxels and then refining local subdomains. This method can rapidly generate optimal material distribution designs for various target shapes, including FE-generated and algorithmically-generated shapes, as well as irregular shapes using patch representation and normal distance loss. The designs are computationally and experimentally validated. The voxel-level inverse design approach empowered by machine learning opens a new paradigm for intelligent design and fabrication of 4D printed active composites, providing new insights for 4D printing applications.

After a thorough review of the entire manuscript, I believe that despite the systematic nature of this work, it does not present significant advancements compared to the author's previous research (DOI: 10.1002/adfm.202109805). In other words, the methods proposed in this paper

have already been introduced in the author's prior publications. Given the journal's emphasis on the novelty of research contributions, I recommend reconsidering the acceptance of this manuscript.

Furthermore, this paper still needs to address the following concerns:

1. Data set generation could consider a broader range of material distribution patterns, not just the two modes of full randomness and island distribution. Incorporating more modes may enhance the model's generalization ability.
2. Evaluating the impact of different boundary conditions on model prediction performance on the test set can be considered, which can further aid in selecting the optimal boundary condition settings.
3. Experimenting with different machine learning model architectures, such as convolutional neural networks, graph neural networks, etc., and comparing them with the current residual network model is an option.
4. It is worth considering validating this design approach on multiple real material systems, not limited to a thermal expansion/contraction material system, to assess the method's applicability.
5. More constraints can be imposed on the target shape, such as surface smoothness constraints, to make the optimization results more in line with practical manufacturing requirements.
6. I noticed that the authors did not use widely adopted open-source deep learning frameworks commonly used in both the research and industry sectors, such as TensorFlow, etc. I have reservations about the reliability of the data obtained in this paper, and the fact that all MATLAB code used in this study has not been made open-source could significantly diminish the paper's impact.

Reviewer #3 (Remarks to the Author):

This paper addresses the challenge of designing active composites (ACs) with 3D shape changes using 4D printing technology. The authors propose an approach that combines machine learning (ML) with gradient-descent (GD) and evolutionary algorithm (EA) techniques for both forward shape prediction and inverse material-distribution design. A residual network ML model is developed for forward shape prediction and a global-subdomain design strategy with ML-GD and ML-EA is used for inverse material distribution design. The ML models enable efficient exploration of the large design space associated with 4D-printed ACs. The results are good, but I have some questions.

1. In this article, you did not say anything about the properties (constitutive model) of the material (from line 901 to line 909). Did you write UMAT for Abaqus when simulation? How can you ensure the material you printed can satisfy the properties you used for the simulation what is "the coefficient of thermal expansion is set to be 0.001 for active material and 0 for passive material. A 50-degree temperature increase is applied to the entire plate"? Even from the authors' previous work (literature 15 and 43), I also didn't find the constitutive model of the materials you use. If it can not be satisfied, this control of the 4D-printed active plate is not precise. You just simulate the trend of the deformation, you didn't know how much this part can deform.
2. In the supplementary videos, all parts are only shown in their final manufactured form, which seems perfect. The deformation process of these parts is not shown at all. Therefore, although it looks perfect, readers will question how these parts that "perfectly" fit the desired shape are "manufactured", which is better if you record the video of the deformation process,
3. For one side of the plate (now a square plate with specified length, width, and thickness), it needs a lot of time (about more than 100 hours) to generate datasets for training. Do you think this is meaningful? or do you have the choice to reduce the time to generate the dataset for training? Also for ML models, it also takes more than 10 hours to train, and the author uses V100 GPU, which is already very powerful. I don't think it will be greatly improved if you use more advanced GPUs like A100 and H100. If you use these more advanced GPUs, it will still take about three or four hours for training.
4. In lines 762 and 763, you said the coefficient of thermal expansion is set to be 0.001 for active material and 0 for passive material. A 50-degree temperature increase is applied to the entire

plate. So, the total expansion ratio will be 0.05 (5%) for the active material. Do you think this tiny change in the volume of active material will cause a huge shape change for the whole structure? I have this doubt due to the authors' previous work in Reference 43. In Figure S1 of supplementary materials in Sun et al. 2022 *Advanced Functional Materials*, such a tiny change in volume of active material (10% in that literature), will it cause more than the 180-degree reverse of that 4D-printed beams??? Even if such a large displacement can occur, during the deformation process, a certain place of the part "hits" the origin on the left side of the part. How do you deal with this situation? This becomes a contact problem in Abaqus simulation, which is very complicated.

5. One tiny issue, in Line 477 " $\geq 2 \times 2$ ", why do you use "x" to represent the multiplication sign? Why don't you express the multiplication sign like in Line 475?

6. What will happen if the target shape is out of the range that the printed part can achieve? For example, if you want the first voxel near (0, 0, 0) to reach the point (1, 1, 1000), this target is impossible to reach. What impact does this situation have on your algorithm?

Reviewer #4 (Remarks to the Author):

This paper by Sun et al. uses machine learning to allow the forward prediction and inverse design of the 3D shape a 2D printed voxelated sheet will take after expansion of specific pattern of voxels defined by the printing process.

The results show that the prediction works very well as arbitrary shapes designed by crumpling a piece of paper can be mapped and printed. It is difficult to comment further on the modeling as this is not my area of expertise.

In terms of manufacturing the use of 4D printing to convert flat sheets into three dimensional objects is of broad interest. Bilayer structures are widely used as analytical predictions of the shapes may be made. However, it much more difficult to print a complex shape with non-periodic features. This work appears to be a significant advance in the design of these types of 4D printed materials.

It is very useful that the modeling is tested by experimental 3D printing of the same materials. It is noted that the modeling considers the active material one that isotropically expands, while in the 3D printed parts the shape change is achieved by contraction of voxels through the evaporation of unreacted monomer. One assumes from the agreement between the experiment and the model that the corresponding 'active' voxel in the experimental material is therefore the well-cured phase. This should be more explicitly indicated in the 'Materials and Methods'.

It would be useful to list the voxel dimensions in the experimentally printed material. Looking at the scale bars in Figure 7, it appears to be roughly on the order of a 1x1x1 mm? Building on this comment, the 3D printing material is interesting and may be useful in some applications, however the evaporation of acrylate monomer may be undesirable in others. If the dimensions at this scale for the voxels, it would be helpful to briefly discuss the applicability of this method to other voxelated printing schemes, perhaps using materials of different thermal expansion to achieve similar effects.

RESPONSE TO REVIEWERS

We appreciate the insightful comments from the four reviewers and sincerely thank them for their time spent on reviewing our manuscript. We have revised our paper according to their suggestions. Below are our point-by-point responses (in blue fonts). The corresponding changes made in the manuscript are highlighted in yellow.

Reviewer #1 (Remarks to the Author):

General comments:

This paper presents the design optimization of pixelated active composite materials, to achieve target deformation behaviors. To reduce the computational cost, the author first trained a ResNet model which performs fast prediction. Materials are then optimized using gradient descent and evolutionary algorithms based on neural network queries. Overall, this research is well presented with results validated by both numerical simulations and real experiments through 3D-printing. Implementation details are provided to allow reproduction. However, the contribution of the manuscript is more on the application of ML-based optimization on deformation optimization of smart composites, rather than the optimization algorithm itself, as similar methods have been seen in many literatures: CT Chen, GX Gu - *Advanced Science*, 2020, and S Lee, Z Zhang, GX Gu - *Materials Horizons*, 2022, etc. So it would be better if the author could elaborate more (in Intro) on the application side for motivation.

Response: Thanks for your suggestion. We agree that our contribution is on applying the ML approach to the shape-change design of active composites (ACs), more specifically, on voxel-level design of 4D-printed AC plates. However, we would like to clarify that although the general idea of ML-based optimization algorithm can be seen in the literature, there exist big gaps between existing methodologies and the current problem (shape-change design) due to some challenges. Besides common challenges (e.g., large design space) in the composite design

problem, the unique challenges in our problem include the highly complex mapping from material distribution to response (large-deflection shape change), the high-dimensional nature of the response data and the variability of the target response (i.e., instead of optimizing or extremizing a single (or a few) property(ies)). These features lead to significantly increased challenge in both accurate forward predictions and efficient inverse designs, and we have developed new strategies/methodologies to address them. For example, our design task is very sensitive to the prediction accuracy, as minor prediction inaccuracies (e.g., deviations in a few data points) can significantly alter the actually achieved shape from the target. We thus dedicated substantial efforts (detailed in our response to your last comment) to enhancing the ML accuracy, ultimately achieving the R-squared > 0.99 . Moreover, new design strategies were developed to achieve the efficient inverse design. These strategies offer insights into the use of ML for other composite shape morphing problems. In addition, our method is generally applicable across various material systems, actuation mechanisms, and length scales. We believe that our work introduces significant advancements in the field of 4D printing by tackling the complex challenges associated with 3D shape morphing of plates.

We have revised the following text in Introduction to emphasize the challenges of shape-change response of ACs (see highlighted text on Page 5):

“For example, Gu and coworkers have made extensive explorations [32-35] on utilizing ML capabilities, such as combining ML with gradient descent (GD) and active learning [32] or with EA [33], for materials design. However, existing works mainly focused on optimizing mechanical properties of materials, such as strength and toughness of composites [34, 35], auxetic metamaterials [36, 37], and responses of soft pneumatic robots [38]; there is limited work on ML-based design of shape changes of ACs [39-43]. Compared to optimizing or extremizing a few properties, the design for shape changes has unique challenges such as highly complex mapping from material distributions to shapes (particularly for large deflections), the high-

dimensional nature of the shape data and the variability of the target shapes. This also places higher demands on the accuracy of ML models. Therefore, there exist big gaps between existing methodologies and the shape-change design of 4D-printed ACs.”

The suggested references have been added to Introduction:

“32.Chen, C.-T. and G.X. Gu, *Generative Deep Neural Networks for Inverse Materials Design Using Backpropagation and Active Learning*. Advanced Science, 2020. 7(5): p. 1902607.

33. Lee, S., Z. Zhang, and G.X. Gu, *Generative machine learning algorithm for lattice structures with superior mechanical properties*. Materials Horizons, 2022. 9(3): p. 952-960.”

We have also added more text to state our motivation on the application side in Introduction (see highlighted text on Page 8):

“Although many ML-based strategies have been utilized for the materials design[32-35], applying existing methodologies to the shape-change design of 4D-printed ACs is very challenging due to the high complexities outlined above.”

There are also a couple of detailed questions regarding the modeling and optimization method used in this work, as below.

Comment 1. The author trained a ResNet as a surrogate model to make fast inferences for different AC designs. The training process seems to be fully supervised. As the model is predicting the displacement (coordinates) of the voxels, is it possible to add in some physical constraint like continuity or smoothness of the deformed composite, which potentially reduce the amount of data needed?

Response: Thanks for your insightful comment regarding the physical constraints. We think that incorporating constraints related to the continuity or smoothness might facilitate the model's training in the initial stages, but might not be able to aid in learning the complex mapping from material distributions to the shape-change predictions. We agree some physics-informed loss functions based on the governing equations might potentially reduce the amount of data needed. However, implementing such an approach presents substantial challenges including the effective integration of appropriate physical laws, which appears to be an entirely new methodology and will be a nice topic for future research. Given these considerations, the strategy to incorporate physical constraint in the model training will be left to our future efforts. This will allow us to dedicate necessary resources and time to thoroughly investigate and explore such methodologies. We have added the following discussion and references (see highlighted text on Page 35):

“Third, our ML model is purely supervised by data. A physics-informed ML model [69, 70] that incorporates appropriate physical constraints into the loss function could potentially reduce the amount of data needed, which will be explored in our future research.”

“69. Raissi, M., P. Perdikaris, and G.E. Karniadakis, Physical_Informed_Deep_Learning Part I Data driven solutions for nonlinear PDEs. 2017.

70. Chen, C.-T. and G.X. Gu, Physics-Informed Deep-Learning For Elasticity: Forward, Inverse, and Mixed Problems. 2023. 10(18): p. 2300439.”

Comment 2. The AC in this work has a design space of size $15 \times 15 \times 2$. However, for most topology optimization problems in literature, the design space is significantly larger, some are like 100×100 or even larger. Is it still viable to train a surrogate model for design optimization when each simulation takes hours to accomplish? It would be great if the author can add in some discussion regarding the scalability of surrogate model based optimization.

Response: Thanks for your suggestion. First, we would like to highlight the advantage of our approach over existing methods for the shape-change design. Most topology optimization (TO) efforts are focused on single material structure; among those of the composite design, only a few works address the shape-change design of 4D-printed ACs. Examples include those by Maute et al. (Refs.18-19), which involve a larger number of design voxels but require significantly longer computational time. For example, the approach in (Ref.18) takes about 50 hours for optimizing a single target. Additionally, these target shapes are typically limited to small-deflection cases (Ref.18) or are defined only by a few feature points (Ref.19), which are relatively less challenging than large-deflection, full-surface-defined targets that add more complexities to TO applications. Moreover, TO faces challenges such as the local minima problem and the complicated derivation of gradients (particularly non-trivial for complex material behaviors), as pointed out by (Ref.32: GX Gu, Advanced Science, 2020). It should be noted that gradient-free EA-based TO frameworks (Refs.22-23) have also been explored for shape-change design, which, however, use smaller numbers of design voxels (e.g., 3×24 voxels in Ref.23) and incur greater time cost (about 12 hours for a single target). Given the context described above, developing new inverse design approaches for shape morphing and 4D printing is highly desired. ML, in particular, holds great promise in tackling challenges of TO as it enables ultrafast and differentiable forward predictions. Our ML-based approach demonstrates highly efficient inverse design for complex target shapes. More importantly, once an accurate ML model is trained, it can be reused for efficient inverse designs of many different target shapes. Therefore, the upfront time cost for obtaining the ML model is offset by the significant time savings of inverse design, when compared to conventional methods as discussed above. In addition, our approach has general applicability across various shape-morphing material systems in 4D printing, which we believe presents significant advancements in the field.

Second, scalability is indeed important in surrogate model-based optimization. In the literature,

ML models concerned with the shape-change response typically use relatively small number of voxels (Gu 2019 ATS; Gu 2022 AS). In our AC plate problem, moving to a larger design space, such as $45 \times 45 \times 2$ voxels, may not lead to (significantly) greater time cost for FE simulations, since the element number ($45 \times 45 \times 4$) of our current FE model may be sufficient. Going even higher to, e.g., $90 \times 90 \times 2$ voxels (assuming $90 \times 90 \times 4$ elements), should still be viable for our problem, since the use of 10 cores of a CPU in our parallelization (10 simulations at a time) does not represent a substantial computational resource. In fact, our current CPU (Intel i9-10900) supports running 20 simulations simultaneously owing to its 10 physical cores with hyper-threading. By using more CPU cores (e.g., 60), which are readily available in many research settings (e.g., across multiple computers or within a cluster), the FE simulation time can be greatly reduced. This suggests the feasibility of scaling our design space without leading to prohibitive time costs. In addition, our approach may be adapted to accommodate the design of large-scale targets without retraining a new ML model. For instance, we can combine multiple conforming patches to construct surfaces containing a larger number of voxels. Such potential modifications to the existing ML model and design approach for AC plates are our ongoing efforts. This idea was demonstrated for AC beams in our recent study (JMPS, doi.org/10.1016/j.jmps.2024.105561). We are currently working on extending this idea to AC plates.

Third, in scenarios where each FE simulation takes a considerably longer time (e.g., hours), the following strategies may be explored and combined: developing smaller-scale ML models with extrapolation capabilities or exploring extrapolation techniques (e.g., splicing); utilizing small datasets with physics-informed loss; exploring data augmentation techniques; exploring reduced-order models for expedited simulations; and conducting parallel computing for expedited simulations. Moreover, it is essential to assess whether increasing the design space is necessary for the design objectives (e.g., desired accuracy and feature size for target approximation). It is

suggested to use a hierarchical concept to progressively increase the design space (e.g., 5×5 to 15×15 to 45×45 for our specific problem), until the design space is sufficient to fulfill the design objective. After all, a larger design space offers enhanced design capabilities but needs more data for model trainings and adds more complexities to optimizations. As the design space increases, the feasibility and the trade-offs involved would require careful consideration for each specific problem.

We have added discussions on the benefits and efficiency gains of our approach for the design process, the potential ways to reducing time cost and its implication on the scalability of our model, and the potential ways to expanding the design space without retraining new models (see highlighted text on Pages 34-35, 38):

Pages 34-35

“In addition, ML allows for rapid shape predictions and efficient gradient computations via AD, thereby enabling the computationally low-cost GD process. This thus offers new possibilities for addressing the challenges faced by TO (which can be seen as FE-GD), the local minima problem and the complicated gradient derivation [32]. More importantly, once an accurate ML model is trained, it can be reused for efficient inverse designs of many different target shapes. Therefore, the time cost for obtaining the ML model is offset by the significant time savings of inverse design, when compared to conventional design methods [18, 22, 23]. Furthermore, both data generation and model training can be further parallelized to improve computational efficiency. For example, in our 10-group parallelized FE simulations, the use of 10 cores of a CPU does not represent a substantial computational resource. By using more CPU cores (e.g., 60), which are readily available across multiple computers or within a cluster, the FE simulation time can be significantly reduced. This also suggests the feasibility of scaling our design space (e.g., to $45 \times 45 \times 2$ or $90 \times 90 \times 2$ voxels) without leading to prohibitive time costs.

.....

For the cases where the initial shape is not square, for example, triangular or rectangular, or those with finer features that cannot be adequately captured by the current design space ($15 \times 15 \times 2$ voxels), our design approach may still be applicable without training a new ML model. For example, one can use cutting if the initial shape is triangular or rectangular, or combine multiple conforming patches to construct surfaces containing a larger number of voxels. These potential modifications could expand the design space of our current ML model and are our ongoing efforts. This idea was demonstrated for AC beams in our recent study [41].”

Page 38

“For the dataset generation, we perform FE simulations in parallel across 10 groups, each using a single core of a CPU (Intel Core i9-10900), which takes about 95 hours. This setup does not represent a substantial computational resource. In fact, our current CPU supports running 20 simulations simultaneously owing to its 10 physical cores with hyper-threading. Therefore, the FE simulation time may be significantly reduced depending on a user’s computational resource.”

Comment 3. From the optimization results in Fig. 5 and 6, the activated shapes indeed match well with the target shape macroscopically. But the deformed materials are not as smooth as the target shape microscopically. Is this problematic in application? How can this issue be potentially addressed?

Response: Thanks for bringing up this question. First, it might not be problematic in most 4D printing applications such as smart devices, actuators, and robotics, where it is more important to achieve the target shape morphing or motion macroscopically. Second, it may be important in certain applications such as deployable reflectors on satellites. To address the problem in this

case, we observe that the surface roughness arises from variations in the curing distance for different grayscale levels of DLP printing, which is reflected in the smoothness of the surface (e.g., both as-printed and actuated plates show the surface roughness as shown in Supplementary Movie 7). This issue could potentially be addressed through the optimization of printing parameters (e.g., light field distribution) for more accurate DLP printing. Previous work of our group has demonstrated the effectiveness of pixel-level manipulation to improve printing accuracy (AFM, doi.org/10.1002/adfm.202213252). The modeling framework developed there can be integrated with a parameter optimization algorithm for smoothing the surface of 4D-printed plates. This is an ongoing project that we are currently pursuing, but it falls outside the scope of the present work.

We have added discussions on the potential improvement of surface smoothness in Section 2.8 (see highlighted text on Page 35):

“Second, our printed AC sheets exhibit some unsmooth features at the voxel boundaries, which arise from variations in the curing distance for different grayscale levels of DLP printing. This issue could potentially be addressed through the optimization of printing parameters [68] to achieve more accurate printing of our optimized material distributions. This is our ongoing work.”

Comment 4. Surrogate model based optimization can be problematic as the optimization process not only maximize/minimize the design objective, but may also converge at someplace with maximized prediction errors. In this study, this phenomenon isn't obvious as the model is trained on a rich dataset and have an extremely high R-squared value. But how would the optimization strategy be improved if the surrogate model gives certain bias from ground truth, say only 85-90% R-squared value?

Response: This is a great question that we have been fighting in our work. Thank you for bringing this up. In our study, we prioritize high accuracy in the forward ML model due to the high sensitivity of our design task—approximating target shapes represented by the high-dimensional data ($16 \times 16 \times 3$ numbers)—where even minor prediction inaccuracies (e.g., deviations in a few data points) can significantly alter the actually achieved shape from the target. To address this, we dedicated substantial efforts (such as generating dataset with distinct types of design patterns, augmenting data using symmetries, optimizing the network architecture, and selecting appropriate boundary conditions) to enhancing our ML model's performance, ultimately achieving the R-squared > 0.99 . Based on our experience in our previous work (1D AC beam) and this work, the high level of accuracy is crucial for our task of shape change design.

In scenarios where a surrogate model exhibits lower accuracy (85-90% R-squared value), as you mentioned, we agree that such models may not meet the stringent requirements of tasks that demand high prediction accuracy, like ours. They could, however, be better suited for tasks less sensitive to accuracy, such as optimizing or extremizing a single property. If the tasks are indeed sensitive to the prediction accuracy but the ML model is not accurate enough, we suggest exploring potential solutions like employing an active learning approach to iteratively refine the model (see your suggested paper, Ref.[32], Chen and Gu, Adv. Sci., 2022) or integrating finite element (FE) verification with ML optimization to validate and adjust predictions (see PNAS, 2023, 120(35): e2309062120). While these strategies could mitigate some of the accuracy concerns, they also introduce additional computational costs. Thus, their feasibility and the trade-offs involved would require careful consideration and further investigation. This is a future work that we are considering.

We have added text to emphasize the importance of high ML accuracy in our design task in Introduction (see highlighted text on Page 5):

“Compared to optimizing or extremizing a few properties, the design for shape changes has unique challenges such as highly complex mapping from material distributions to shapes (particularly for large deflections), the high-dimensional nature of the shape data and the variability of the target shapes. This also places higher demands on the accuracy of ML models.”

Reviewer #2 (Remarks to the Author):**General comments:**

This paper presents a machine learning and evolutionary algorithm enabled approach for designing and predicting shape changes of 4D printed active composite plates. A residual network based machine learning model is developed to predict shape changes with great accuracy and efficiency. The model is then combined with gradient descent and evolutionary algorithms to achieve inverse design of optimal material distributions for desired shape changes, which is significantly accelerated by machine learning. A global-subdomain design strategy is proposed to obtain optimal designs by first globally optimizing all voxels and then refining local subdomains. This method can rapidly generate optimal material distribution designs for various target shapes, including FE-generated and algorithmically-generated shapes, as well as irregular shapes using patch representation and normal distance loss. The designs are computationally and experimentally validated. The voxel-level inverse design approach empowered by machine learning opens a new paradigm for intelligent design and fabrication of 4D printed active composites, providing new insights for 4D printing applications.

Response 1 to general comments: We thank you for your thorough review of our paper and have made substantial revisions to the manuscript.

After a thorough review of the entire manuscript, I believe that despite the systematic nature of this work, it does not present significant advancements compared to the author's previous research (DOI: 10.1002/adfm.202109805). In other words, the methods proposed in this paper have already been introduced in the author's prior publications. Given the journal's emphasis on the novelty of research contributions, I recommend reconsidering the acceptance of this manuscript.

Response 2 to general comments: We respectfully disagree with your assessment regarding the novelty and significance of this work compared to our previous research (Ref.40, AFM, 2022). While our earlier work focused on the 2D shape change of active composite (AC) beams, the current work advances into the 3D shape change of AC plates, representing a significant leap in both complexity and application. The transition from 2D to 3D involves addressing some new significant challenges, developing innovative strategies, and ultimately impacting a broader range of applications in 4D printing. Below, we detail the key aspects of our response:

Challenges: The transition from beams to plates introduces significantly higher physical and data complexities, presenting considerable challenges in both accurate ML prediction and efficient inverse design. For example, for an AC beam design using 24×4 voxels for 2D shape change, the design space is ($2^{96} \approx 8 \times 10^{28}$); for an active plate with $15 \times 15 \times 2$ voxels, the design space becomes 2^{450} ($\approx 3 \times 10^{135}$), an increase of more than 10^{100} . There thus exist big gaps between our previous methodologies (ML and evolutionary algorithm, ML-EA) and the current problem, which requires developing new methodologies.

Strategies/efforts: To address the challenges, our new methodologies are detailed as follows. First, we did a comprehensive exploration of strategies to enhance ML model accuracy, such as investigating boundary conditions, data augmentation, network architectures, and symmetry-enabled accuracy enhancement. These efforts have been expanded during revision, including broadening dataset ranges and experimenting with new ML model architectures. Second, we developed a novel global-subdomain design strategy, inspired by the spatially sequential dependency in plate deflection. We also explored a machine learning and gradient descent (ML-GD) approach tailored for our discrete-variable problem, which, combined with subdomain EA, surpasses the efficiency of our previous ML-EA strategy.

Impact/Significance: The ability to design shape-morphing plates has profound implications across a wide spectrum of 4D printing applications, from smart devices and robotics to biomedical devices. Our work represents a substantial progress in the field and provides an intelligent design-fabrication paradigm for 4D printing. Our methodologies/explorations offer valuable insights into addressing problems related to the design for AC plates, shape morphing, and 4D printing. In addition, our approach demonstrates general applicability across various material systems, activating stimuli, and length scales, further showing the novelty and broader relevance of our current study.

In summary, our work introduces significant advancements in the field of 4D printing by tackling the complex challenges in designs for 3D shape morphing of plates. The novel strategies and notable impact of our work strongly align with the journal's emphasis on original research contributions.

Furthermore, this paper still needs to address the following concerns:

Comment 1. Data set generation could consider a broader range of material distribution patterns, not just the two modes of full randomness and island distribution. Incorporating more modes may enhance the model's generalization ability.

Response: Thanks for your valuable suggestion. We have expanded our dataset beyond the original fully random and island patterns to include additional datasets with new pattern types.

First, we introduced a hierarchical pattern type inspired by the Sci. Adv. paper by Buehler and colleagues. These hierarchical patterns utilize "super" voxels composed of $i \times j$ ($i, j \geq 2$) true voxels, creating an $m \times n$ hierarchical grid for generating random designs. To cover a broad range

of patterns, we developed a dataset with mixed hierarchies (named "HrchMix") by randomly sampling design hierarchies (m and n) from uniform distributions ($m, n \sim U(3, 7)$), followed by the generation of random grids ($\sum i = \sum j = 15$), random combination of random grids, and finally random assignment of two materials. The two layers of each material distribution can have different hierarchies to ensure diversity. In addition, we generate two independent validation datasets with the 3×3 and 5×5 hierarchies, where each "super" voxel contains 5×5 and 3×3 true voxels, respectively. Representative patterns for the HrchMix, 3×3 , and 5×5 datasets are illustrated in **Figure R1**. We also study their statistics and the results are shown in **Figure R2**.

Second, we explored a completely different pattern type inspired by Spinodal decomposition (Ref.5 of Supplementary Materials), a process related to phase separation that results in distinctive patterns useful for material design. To cover a broad range of patterns, the Spinodal dataset (named "Spnd") was generated with varying anisotropies, orientations, and periods, details of which are provided in the newly added Section S5 of Supplementary Materials. Representative patterns for the Spnd datasets are illustrated in **Figure R1**. We also study their statistics and the results are shown in **Figure R2**.

The representative patterns and statistics of these new datasets confirm that they differ from the existing fully random and island datasets. We split the HrchMix and Spnd datasets into training and validation sets and incorporate the new training sets into our original training set, maintaining the overall size by replacing an equivalent number of original datapoints. The combined training set was then used to train a new ResNet-based ML model, the performance of which, compared to the existing model across various validation sets (original, HrchMix, Spnd, 3×3 and 5×5), is shown in **Figure R3**. First, our existing ML model shows an excellent performance on the new datasets, demonstrating its strong ability to generalize. Second, incorporating these new dataset types into the training set improves the model's performance on

the independent 3×3 and 5×5 validation sets, indeed indicating an enhancement in generalization ability, although the improvement in prediction accuracy is not significant (this is probably our existing model already has high accuracy). Due to the slight improvement of the new model and the excellent performance of the existing model, we opted not to rerun the inverse designs with the new model. We have added a new Section S5 in Supplementary Materials for implementation details of the new datasets. We have also added the following descriptions and results in main text (see highlighted text on Page 16):

“Moreover, to evaluate the model’s generalization ability, we build additional datasets that differ significantly in pattern types from the existing fully random and island datasets, as detailed in Supplementary Materials. The distinction across datasets is illustrated through their example patterns (**Figure S6**) and statistics (**Figure S7**). Our ML model exhibits excellent performance on the new datasets, and incorporating the new datasets into the training set slightly improves the performance (**Figure S8**). These demonstrate that our existing ML model has strong generalization ability.”

Figure R1. (Figure S6). Representative patterns for the hierarchical patterns and spinodoid patterns.

Figure R2. (Figure S7). The statistics of the newly generated datasets. We also calculate the maximum displacement of each datapoint (design-shape pair) of the entire dataset and then depict the distributions of the maximum displacement for (a) HrchMix dataset, (b) 3×3 and 5×5 datasets, (c) Spnd1 dataset generated with isotropic spinodoid patterns and Spnd2 dataset generated with anisotropic spinodoid patterns, and (d) Spnd dataset dataset. All datasets are based on the converted BCs. The mean value of the maximum displacements of each dataset is shown in the corresponding panel. These new datasets overall exhibit a maximum displacement between that of the fully random dataset and that of the island dataset.

Figure R3. (Figure S8). Effects of new dataset types on the ML model’s performance. (a) Performance of two ML models on different validation sets in terms of the R^2 value. (b) Performance of two ML models on two independent validation sets.

Comment 2. Evaluating the impact of different boundary conditions on model prediction performance on the test set can be considered, which can further aid in selecting the optimal boundary condition settings.

Response: Thanks for your helpful suggestion. We have accordingly differentiated our response based on two distinct categories of BCs. The first category of BCs allows for the free shape morphing of plates. This category, which is our primary focus, is pivotal to the majority of 4D printing applications, where different BCs essentially give the same shape changes but with different rotations and thus different coordinate representations. Following your suggestion, during the revision, we have considered two additional BCs, namely BC3 and BC4, alongside the original and converted BCs previously considered. The forms of new BCs are provided in the newly added Section S1 of Supplementary Materials, and their effects on the model performance are shown in **Figure R4**. The deep model (ResNet-33) shows greatly improved performance with the converted BCs compared to the original, BC3, and BC4; the latter three show similar performance. This observation suggests that retaining the spatially sequential dependency in plate morphing can significantly enhance prediction accuracy. To further verify this, we compare the ML (LSTM) performance on the beam's shape morphing under two BCs, simply supported and cantilever-like, which again yield identical shape changes but differ in coordinate representation. **Figure R5** shows that transforming shapes from simply-supported to cantilever-like BCs enables a substantial accuracy improvement, underscoring the ease of learning for shapes with sequential dependency by ML models.

The second category of BCs involves boundary constraints or applied mechanical loads that can interfere the shape morphing of plates. While our primary focus is on the first category, the second category can be relevant to certain applications, which would need new ML models. Further, a universal ML model capable of accommodating general BCs or loads would be highly beneficial. Due to the increased complexity of such a problem and the fact that it is beyond the scope of the present paper, new ML models on it will be explored in future studies.

We have added **Figure R4** as **Figure 3c** and revised the relevant text in main text (see

highlighted text on Pages 10 and 14). We have also added a new **Section S1 in Supplementary Materials**. Our evaluation of different BCs can provide insights into optimizing BC settings for enhanced ML performance in other shape morphing problems (e.g., those with different structures and mechanisms).

Figure R4. (Figure 3c). Boundary condition (BC) effects. Validation loss versus epochs showing the training progress of ResNets using four datasets with different BCs: original BCs (oriBC), converted BCs, BC3, and BC4.

Figure R5. ML (LSTM) model performance on the beam shape-change problem with two boundary conditions: (a) Simply-supported BC, (b) cantilever-like BC converted from the simply-supported BC, where B' is right point of the first available element on the beam mid-axis. By converting shapes from simply-supported to cantilever-like BC, a substantial accuracy

improvement is achieved for the same ML architecture.

Comment 3. Experimenting with different machine learning model architectures, such as convolutional neural networks, graph neural networks, etc., and comparing them with the current residual network model is an option.

Response: Thanks for your helpful suggestion. We experimented with CNNs of different depths and the results were compared with ResNets in **Figure 3**. In response to your suggestion, we have extended our investigation to include graph convolutional networks (GCNs) (**Figure R6a**). We studied GCN models with different depths and architectures to ensure a fair comparison. Details on the GCNs tested are provided in the **newly added Section S4 of Supplementary Materials**. As shown in **Figure R6b**, the GCN-21×64 (depth × hidden size) with 4 skip connections (SCs) is the top performer among the GCNs tested. However, its validation MSE is higher (indicating lower accuracy) than that of the best-performing CNN and ResNet models (see **Figure 3d**). Further, the regression plots for the best-performing GCN and CNN models are shown in **Figure R6c** and **R6d**, respectively, revealing that GCNs exhibit inferior performance compared to CNNs, and consequently, to ResNets as well (see **Figure 3h**).

Figure R6. (Figure S5). Effects of model architectures. (a) Schematics of GCN architectures. GCL: graph convolutional layer. SC: skip connection. (b) Validation loss versus epochs showing the training progress of GCN with different depths, hidden sizes, and number of skip connections (SCs). GCNII is an extension of GCN with initial residual and identity mapping (Ref.3 of Supplementary Materials). (c-d) Density scatter plots of the true versus ML-predicted coordinate z using the (c) GCN and (d) CNN with optimal hyperparameters.

In addition to these experiments, we also conducted preliminary tests at the beginning of this project using a smaller design space ($5 \times 5 \times 2$) to compare CNNs with other architectures such as LSTM networks and multi-dimensional LSTM (MDLSTM). The LSTM model sequentially predicts the shape changes at increasing active strains (0.01, 0.02, ..., 0.05), and the final prediction corresponds to that by CNN. The MDLSTM model, as an extension of our previous

work (Ref.40, AFM, 2022), treats the x and y dimensions as two “sequential” directions, and sequentially predicts the coordinates column-by-column (each column has two voxels in z -dimension) from one corner to the opposite corner of the plate. Our tests showed that CNNs outperformed the other architectures, further reinforcing our decision to utilize the ResNet architecture for the current design space ($15 \times 15 \times 2$).

It should be noted that the higher performance of CNN and ResNet are attributed to the structured nature of our voxel-level input and output data which aligns well with the convolutional filter’s capabilities. When dealing with non-structured data, such as structures with irregular geometries or topologies, GCNs might be appropriate. This will be explored in future studies.

We have added a new **Section S4 in Supplementary Materials** for the implementation and results of GCN. We have also added relevant descriptions in the main text (see **highlighted text on Pages 13, 15, and 40**).

Page 13:

“In addition to the ResNet, we also build the plain CNN ..., **as well as the graph convolutional network (GCN)**, and compare their performances on the active plate design problem.”

Page 15:

“**In addition, we also study the performance of graph convolutional network (GCN) models, which proves to be inferior to that of both plain CNN and ResNet (Figure S5).**”

Page 40:

“**In addition to ResNet, we also build the plain CNN and GCN models. Our CNN has the same architecture as ResNet but without skip connections. Details on our GCN implementation are provided in Supplementary Materials.**”

Comment 4. It is worth considering validating this design approach on multiple real material systems, not limited to a thermal expansion/contraction material system, to assess the method's applicability.

Response: Thanks for your valuable suggestion. Our design approach is generally applicable to any active composite (AC) structure that exhibits differential dimensional changes driven by mismatched strains due to isotropic volume change under proper activation stimuli. In response, we have validated our approach with two additional grayscale DLP printable material systems, employing different actuation mechanisms, as shown in the newly added Supplementary Movie 6. The successful realization of target shape changes in both systems demonstrates the applicability not only across different material systems but also under varied activation stimuli. In addition, our design approach is applicable across different length scales, which is demonstrated on a smaller AC plate using the same material system, as shown in the newly added Supplementary Movie 5.

Comment 5. More constraints can be imposed on the target shape, such as surface smoothness constraints, to make the optimization results more in line with practical manufacturing requirements.

Response: Thanks for your suggestion. We are not fully clear about the “surface smoothness constraints”, “imposed on the target shape”. Our response is organized into two possible interpretations.

In cases you are talking about the unsmoothness of the target shape, we have introduced a pre-smoothing function during revision (detailed in the newly added Section S11 of Supplementary

Materials), which removes severely unsmooth features of the target, as shown in **Figure R7**. We would like to mention that even without pre-smoothing, our current approach with the normal distance-based loss (Eq.(13)) can implicitly do the smoothing during the optimization. This is because it handles unsmoothed targets by encouraging achieved points to conform to the overall target surface, rather than matching each specific target point. As illustrated in **Figures 7c**, for the crumpled paper, our approach achieves a smoother shape that best approximates the target. This contrasts with the pre-smoothing method discussed above, yet both approaches yield a similar smoothing effect. We have a **new Section S11 in Supplementary Materials** and the following in main text (see **highlighted text on Page 33**):

“Note that for targets with severe unsmoothness, one can pre-smooth the target before the design, as shown in **Figure S13**. Here, without pre-smoothing, our algorithm implicitly smooths the target during the design.”

Figure R7. (Figure S13). The crumpled paper target shape with the severe unsmooth feature being intentionally introduced (top), the smoothed target shape with the unsmoothness being removed (bottom left), and the further smoothed target shape (bottom right).

In cases you are talking about the surface roughness that occurs at the boundaries between different material phases, our response is as follows. We observe that the surface roughness arises from variations in the curing distance for different grayscale levels of DLP printing, which is reflected in the smoothness of the surface (e.g., both as-printed and actuated plates show the surface roughness as shown in Supplementary Movie 7). This issue could potentially be addressed through the optimization of printing parameters (e.g., light field distribution) for more accurate DLP printing. Previous work of our group has demonstrated the effectiveness of pixel-level manipulation to improve printing accuracy (AFM, doi.org/10.1002/adfm.202213252, newly added Ref. 68). The modeling framework developed there can be integrated with a parameter optimization algorithm for smoothing the surface of 4D-printed plates. This is an ongoing project that we are currently pursuing, but it falls outside the scope of the present work. We have added discussions on the potential improvement of surface smoothness in Section 2.8 (see highlighted text on Page 35):

“Second, our printed AC sheets exhibit some unsmooth features at the voxel boundaries, which arise from variations in the curing distance for different grayscale levels of DLP printing. This issue could potentially be addressed through the optimization of printing parameters [68] to achieve more accurate printing of our optimized material distributions. This is our ongoing work.”

“68. Montgomery, S.M., et al., *Pixel-Level Grayscale Manipulation to Improve Accuracy in Digital Light Processing 3D Printing*. *Advanced Functional Materials*, 2023. **33**(17): p. 2213252.”

Comment 6. I noticed that the authors did not use widely adopted open-source deep learning frameworks commonly used in both the research and industry sectors, such as TensorFlow, etc. I have reservations about the reliability of the data obtained in this paper, and the fact that all MATLAB code used in this study has not been made open-source could significantly diminish the paper’s impact.

Response: Thank you for bringing up this point. We choose MATLAB because of its comprehensive documentation and ease of rapidly implementing ideas, which significantly expedited our development process. We understand your concern regarding the reliability of the data obtained; however, we would like to emphasize that the choice of software does not inherently affect the reliability of our results and conclusions. In addition, all ML predictions and optimized designs are verified by FE simulations and experiments, which demonstrates the validity of MATLAB and our research findings.

Regarding the availability of our code, we agree that the open-sourced codes will enhance the impact and reproducibility of our research. To this end, we have already made parts of our work available to the public, including the dataset, the machine learning model, and the inverse design code (see highlighted text on Pages 43-44). Due to time constraints, not all codes have been released yet, but we are committed to open-sourcing the remaining codes as soon as possible.

“Data availability

The generated dataset is available in:

<https://www.kaggle.com/datasets/sunxiaohao/dataset-for-active-shapes-of-ac-plates>

Code availability

The codes are available in:

[https://github.com/XiaohaoSun/ML_4DP_AC_plates/”](https://github.com/XiaohaoSun/ML_4DP_AC_plates/)

Reviewer #3 (Remarks to the Author):

General comments:

This paper addresses the challenge of designing active composites (ACs) with 3D shape changes using 4D printing technology. The authors propose an approach that combines machine learning (ML) with gradient-descent (GD) and evolutionary algorithm (EA) techniques for both forward shape prediction and inverse material-distribution design. A residual network ML model is developed for forward shape prediction and a global-subdomain design strategy with ML-GD and ML-EA is used for inverse material distribution design. The ML models enable efficient exploration of the large design space associated with 4D-printed ACs. The results are good, but I have some questions.

Response: We appreciate your efforts in reviewing our work and would like to address your comments as follows.

Comment 1. In this article, you did not say anything about the properties (constitutive model) of the material (from line 901 to line 909). Did you write UMAT for Abaqus when simulation? How can you ensure the material you printed can satisfy the properties you used for the simulation what is "the coefficient of thermal expansion is set to be 0.001 for active material and 0 for passive material. A 50-degree temperature increase is applied to the entire plate"? Even from the authors' previous work (literature 15 and 43), I also didn't find the constitutive model of the materials you use. If it can not be satisfied, this control of the 4D-printed active plate is not precise. You just simulate the trend of the deformation, you didn't know how much this part can deform.

Response: Thank you for this important question. In our FE simulations, we use the neo-Hookean model with identical moduli for both active and passive phases, and the isotropic strain

mismatch is 0.05 (applied through thermal expansion). The text referenced by you is for the FE model. In experiments, the printed phases were measured to have a modulus ratio of 0.053 and an active strain mismatch of 0.057. While the material properties differ between the FE model and experiments, we employ a multilayer composite bending theory to approximately, quantitatively tune the dimensions of the printed part to control its curvature, thus beyond merely simulating deformation trends. The details are provided in the newly added Section S9 of Supplementary Materials (which is too long and thus omitted here). Specifically, we tune the dimensions of printed parts such that they lead to curvatures close to the FE (or ML) predictions while satisfying the manufacturability condition. This theory's validity is demonstrated in Figure R8, which presents a comparison between analytical curvatures across varying plate dimensions. As discussed in Section S9, this dimension modification approach can approximately compensate for effects of the property difference between the FE model and experiments on the shape-change prediction, thus offering an efficient way to applying our ML model across different materials (thus having different properties) and length scales. It should be noted that our design approach will also work by retraining a new machine learning (ML) model based on practical material properties (expansion mismatch and modulus difference). However, the dimension modification approach can streamline the process and achieve a broader applicability. We thus achieve quantitative control over the deformation of 4D-printed active plates.

Moreover, our previous work (AC beam, now Ref.40, AFM, 2022), where a different material system was used, also demonstrates the effectiveness of this theoretical framework on the multilayer beam system with 4×24 voxels. There, the printed materials have a modulus ratio of 0.15 and mismatch strain of 0.072 between the two printed phases. The analytical model allows us to convert the optimized design patterns to ensure the actual curvatures, derived from experimental parameters, closely matches with the simulated curvatures based on theoretical parameters. This further supports the validity of our dimension modification approach.

Figure R8. (Figure S12). (a) Schematic of a multi-layer bi-phase beam with a strain mismatch (left). Bending of a bilayer beam induced by the strain mismatch (right). (b) Comparison of theory and experiment for the relation between curvature and thickness ratio. (c) Comparison of theory and experiment for the relation between curvature and total thickness. (d) Schematic of a bilayer plate with a strain mismatch.

We have a new Section S9 in Supplementary Materials and the following in main text (see highlighted text on Page 42):

“Note that the material properties in experiments are different from those used in FE simulations. The printed two material phases show a modulus ratio of 0.053 and a strain mismatch of 0.057, while the FE (or ML) model assumes the identical modulus and a strain mismatch of 0.05. Such a property difference would result in different shape changes between ML predictions and 4D-printed parts. This issue can be resolved by retraining a new ML model based on practical

material parameters and rerunning the design. Here, instead of retraining a new model, we adopt a strategy similar to that of our previous work[40, 41], i.e., tune the print dimension to approximately compensate for the effect of property differences through an analytical model for the local curvature. Details are provided in Section S9 of Supplementary Materials. This dimension modification strategy offers an efficient way to applying our ML model across different materials and length scales.”

Comment 2. In the supplementary videos, all parts are only shown in their final manufactured form, which seems perfect. The deformation process of these parts is not shown at all. Therefore, although it looks perfect, readers will question how these parts that "perfectly" fit the desired shape are "manufactured", which is better if you record the video of the deformation process,

Response: Thank you for your helpful suggestion. We acknowledge the importance of showing not only the final shape but also the manufacturing and deformation processes to provide a comprehensive understanding of our work. In response, using the crumpled paper as an example, we have recorded a detailed video (see **Supplementary Movie 7**) capturing a complete process involving both the printing and the subsequent shape-morphing. The printed sheet is initially flat. Through an evaporation process at 80°C, these sheets undergo a transformation, morphing into the desired shapes. For the same example, the process of crumpling a paper, the identified target shape, and the optimized design and shape by our approach, are also shown in the video. We believe the video will not only facilitate the readers’ understanding of how the sheet with desired shape is 4D printed, but also better demonstrate the new paradigm for the intelligent design and fabrication of 4D printing, as presented in this study.

Based on suggestion, we have added the following text.

Page 31 (Caption of **Figure 7**):

“A complete design-fabrication process including the paper crumpling, target identification, inverse design, 3D printing, actuation (shape morphing), and final 4D-printed shape is shown in **Supplementary Movie 7.**”

Page 33 (Section 2.7):

“This movie also shows a complete design-fabrication process from the paper crumpling to final 4D-printed shape.”

Comment 3. For one side of the plate (now a square plate with specified length, width, and thickness), it needs a lot of time (about more than 100 hours) to generate datasets for training. Do you think this is meaningful? or do you have the choice to reduce the time to generate the dataset for training? Also for ML models, it also takes more than 10 hours to train, and the author uses V100 GPU, which is already very powerful. I don't think it will be greatly improved if you use more advanced GPUs like A100 and H100. If you use these more advanced GPUs, it will still take about three or four hours for training.

Response: Thank you for bringing up this important question. First, the ML model is meaningful and beneficial for practical 4D printing design applications. Once the model is trained, it can be reused for efficient inverse designs of numerous target shapes. Thus, the upfront time cost (about 105 hours) of obtaining the ML model is offset by the significant time savings of inverse design, when compared to conventional design methods. For example, in TO frameworks for the shape-change design of ACs (Refs.18-19), the optimization does not need the time of training, but is much slower in the design stage (e.g., about 50 hours for a single target shape in Ref.18), and the targets are less challenging than the large-deflection, full-surface-defined target surfaces in our

study. As another example, existing EA-based TO frameworks (Refs.22-23) are also much slower: optimizing a beam structure composed of much fewer (3×24) voxels takes about 12 hours for a single target shape (Ref.23). Here, our design approach demonstrates high efficiency for complicated target shapes and has general applicability to various 4D printing or shape morphing material systems.

Second, the time needed to obtain an ML model, which involves the dataset generation (about 95 hours) and model training (about 10 hours), can indeed be substantially reduced by leveraging parallel computing. Regarding the dataset generation, the time is mostly spent on finite element (FE) simulations, which, in our case, were conducted in parallel across 10 groups using 10 cores of a CPU. This setup does not represent a substantial computational resource. In fact, our current CPU (Intel i9-10900) supports running 20 simulations simultaneously owing to its 10 physical cores with hyper-threading. By using more CPU cores (e.g., 60), which are readily available across multiple computers or in workstation or cluster environments, the FE simulation time can be significantly reduced. Similarly, the model training was performed on a single V100 GPU. We agree that using more advanced GPUs like A100 and H100 may not offer significant improvement in time cost. However, the significant time reduction could be achieved by using multiple GPUs, a strategy that is increasingly feasible and not difficult to implement in many research settings (e.g., in a cluster environment). As a reference, a recent study (Nat. Mach. Intell., 2023, 5, 1466–1475) employed 8 Quadro RTX 6000 GPUs for 70 hours of training. We can anticipate that adopting similar computational power would substantially shorten our model training time.

Third, our approach may be adapted to accommodate the design of targets with different initial geometries (i.e., not limited to a square plate considered here) without training a new ML model. For instance, we can use cutting techniques in cases where the initial shape is triangular or

rectangular, or combine multiple conforming patches to construct larger surfaces. These potential modifications to the existing ML model and design approach for AC plates are our ongoing efforts. This idea was demonstrated for AC beams in our recent study (JMPS, 2024, doi.org/10.1016/j.jmps.2024.105561). We are also currently working on implementing this to AC plates.

We have added discussions on the time issue, the benefits and efficiency gains of our approach for the design process, and the potential ways to reducing time cost through parallelization (see highlighted text on Pages 34-35, 38):

Pages 34-35

“In addition, ML allows for rapid shape predictions and efficient gradient computations via AD, thereby enabling the computationally low-cost GD process. This thus offers new possibilities for addressing the challenges faced by TO (which can be seen as FE-GD), the local minima problem and the complicated gradient derivation [32]. More importantly, once an accurate ML model is trained, it can be reused for efficient inverse designs of many different target shapes. Therefore, the time cost for obtaining the ML model is offset by the significant time savings of inverse design, when compared to conventional design methods [18, 22, 23]. Furthermore, both data generation and model training can be further parallelized to improve computational efficiency. For example, in our 10-group parallelized FE simulations, the use of 10 cores of a CPU does not represent a substantial computational resource. By using more CPU cores (e.g., 60), which are readily available across multiple computers or within a cluster, the FE simulation time can be significantly reduced. This also suggests the feasibility of scaling our design space (e.g., to $45 \times 45 \times 2$ or $90 \times 90 \times 2$ voxels) without leading to prohibitive time costs.

.....

For the cases where the initial shape is not square, for example, triangular or rectangular, or those with finer features that cannot be adequately captured by the current design space ($15 \times 15 \times 2$

voxels), our design approach may still be applicable without training a new ML model. For example, one can use cutting if the initial shape is triangular or rectangular, or combine multiple conforming patches to construct surfaces containing a larger number of voxels. These potential modifications could expand the design space of our current ML model and are our ongoing efforts. This idea was demonstrated for AC beams in our recent study [41].”

Page 38

“For the dataset generation, we perform FE simulations in parallel across 10 groups, each using a single core of a CPU (Intel Core i9-10900), which takes about 95 hours. This setup does not represent a substantial computational resource. In fact, our current CPU supports running 20 simulations simultaneously owing to its 10 physical cores with hyper-threading. Therefore, the FE simulation time may be significantly reduced depending on a user’s computational resource.”

Comment 4. In lines 762 and 763, you said the coefficient of thermal expansion is set to be 0.001 for active material and 0 for passive material. A 50-degree temperature increase is applied to the entire plate. So, the total expansion ratio will be 0.05 (5%) for the active material. Do you think this tiny change in the volume of active material will cause a huge shape change for the whole structure? I have this doubt due to the authors' previous work in Reference 43. In Figure S1 of supplementary materials in Sun et al. 2022 *Advanced Functional Materials*, such a tiny change in volume of active material (10% in that literature), will it cause more than the 180-degree reverse of that 4D-printed beams??? Even if such a large displacement can occur, during the deformation process, a certain place of the part "hits" the origin on the left side of the part. How do you deal with this situation? This becomes a contact problem in Abaqus simulation, which is very complicated.

Response: Thank you for your interesting questions. First, the expansion strain of 0.05 indeed results in significant shape changes in the structure, as observed in our study. This effect is comparable to our previous work (now Ref.40, Sun et al. 2022 Adv. Funct. Mater.), where a strain of 0.1 was shown to cause extensive deformation, including more than a 180-degree reversal in beam structures, as depicted in Figure S1 of Ref.40. In fact, the beam can morph to loops under a strain mismatch of 0.1 (as the contact is neglected in our FE model). For example, using the bilayer beam theory (Eq.(7) of Supplementary Materials), the curvature of a bilayer with identical modulus and strain mismatch of 0.1 is calculated to be 0.15 mm^{-1} (radius of curvature of about 6.7 mm). As the beam length is 80 mm, it can easily achieve a 180-degree reverse. In the current work on plates, a strain mismatch of 0.05 can thus achieve large shape changes (e.g., 90-degree deflection). It is important to note that the magnitude of shape change is highly dependent on the material distribution pattern, which can be seen from the variations in averaged maximum displacement across different datasets—fully random, island, hierarchical, and spinodal—as detailed in our response to Comment 1 of Reviewer 2.

Second, regarding the contact problem during shape morphing, our simulations did not account for the potential self-contact of the plate, thus allowing for penetration within the model. This assumption is justified for two reasons: (1) only a small subset of the dataset exhibited shape changes significant enough to potentially cause contact issues, and (2) our primary focus is on free shape morphing, which is most relevant to the majority of 4D printing applications. While we recognize the importance of considering contact in certain scenarios, the complexity of accurately simulating contact in FE models led us to prioritize tractability and relevance to the intended application scope of our study. The contact effect and its applications in certain scenarios will be explored in our future work.

We have added text on the neglect of contact effects in Section 4.1 (see **highlighted text on**

Page 37):

“During the shape change, the potential self-contact of the plate is not accounted for.”

Comment 5. One tiny issue, in Line 477 " $\geq 2 \times 2$ ", why do you use "x" to represent the multiplication sign? Why don't you express the multiplication sign like in Line 475?

Response: We thank the reviewer for pointing it out. This is a typo and we have corrected “x” to “ \times ”.

Comment 6. What will happen if the target shape is out of the range that the printed part can achieve? For example, if you want the first voxel near (0, 0, 0) to reach the point (1, 1, 1000), this target is impossible to reach. What impact does this situation have on your algorithm?

Response: Thanks for this interesting question. The algorithm behavior depends on the loss function when the target shape is beyond the achievable range of the printed part. We have adopted two types of loss functions: (1) the weighted mean squared error (MSE) between achieved points and target points (see Eqs.(6) and (11)), and (2) the weighted mean normal distance between achieved points and target surface patches (see Eq.(13)). The response to an unattainable target varies depending on the specific scenario.

First, if the entire target is out of reach, such as all points near (1, 1, 1000), the algorithm would produce a shape that deflects towards these target points. Since we always preprocess the target shape such as adopting appropriate boundary conditions and scaling the target surface dimension, we are unlikely to encounter a completely unattainable target.

Second, in cases where one target point is significantly out of reach while others are attainable, the result becomes more complex, akin to dealing with a target having severe unsmoothness or kinks. As the algorithm is to minimize the loss, the achieved shape is likely to be random, depending on the overall target profile. To improve handling of such scenarios, we have introduced a pre-smoothing function during revision (detailed in the newly added Section S11 of Supplementary Materials), which removes severely unsmooth features of the target, as shown in **Figure R9**.

Third, for targets with moderate local unsmoothness or kinks, our existing algorithm setup, particularly the second loss function (Eq.(13)), can implicitly do the smoothing without the pre-smoothing. This loss function allows for robust handling of irregular targets by encouraging achieved points to conform to the overall target surface, rather than matching each specific target point, which is illustrated with unsmoothed target shapes in **Figures 7c** and those with irregular boundary in **Figure 7d**. Thus, for unsmoothed targets, the second loss function achieves a smoother shape that best approximates the target. This contrasts with the pre-smoothing method discussed above, yet both approaches yield a similar smoothing effect.

We have a new Section S11 in Supplementary Materials and the following in main text (see highlighted text on Page 33):

“Note that for targets with severe unsmoothness, one can pre-smooth the target before the design, as shown in **Figure S13**. Here, without pre-smoothing, our algorithm implicitly smooths the target during the design.”

Figure R9. (Figure R7, Figure S13). The crumpled paper target shape with the severe unsmooth feature being intentionally introduced (top), the smoothed target shape with the unsmoothness being removed (bottom left), and the further smoothed target shape (bottom right).

Reviewer #4 (Remarks to the Author):

General comments: This paper by Sun et al. uses machine learning to allow the forward prediction and inverse design of the 3D shape a 2D printed voxelated sheet will take after expansion of specific pattern of voxels defined by the printing process.

The results show that the prediction works very well as arbitrary shapes designed by crumpling a piece of paper can be mapped and printed. It is difficult to comment further on the modeling as this is not my area of expertise.

In terms of manufacturing the use of 4D printing to convert flat sheets into three dimensional objects is of broad interest. Bilayer structures are widely used as analytical predictions of the shapes may be made. However, it much more difficult to print a complex shape with non-periodic features. This work appears to be a significant advance in the design of these types of 4D printed materials.

Response: We appreciate your positive feedback and would like to address your comments as follows.

Comment 1. It is very useful that the modeling is tested by experimental 3D printing of the same materials. It is noted that the modeling considers the active material one that isotropically expands, while in the 3D printed parts the shape change is achieved by contraction of voxels through the evaporation of unreacted monomer. One assumes from the agreement between the experiment and the model that the corresponding ‘active’ voxel in the experimental material is therefore the well-cured phase. This should be more explicitly indicated in the ‘Materials and Methods’.

Response: Thanks for your valuable suggestion. We have revised the following text in Section 4.5 to explicitly state what active and passive phases correspond to in the experiments (see highlighted text on Page 41):

“Our ML optimal designs are converted into grayscale printing slices such that the active (“1”) and passive (“0”) voxels are printed using brighter (0% grayscale) and dimmer (60% grayscale) lights and thus lead to well-cured (higher-DoC) and partially-cured (lower-DoC) material phases, respectively. The printed structure is then heated to facilitate the monomer volatilization. The partially-cured phase (“0”) contains more residual monomers that can volatilize at elevated temperatures and thus shows more volume shrinkage than the well-cured phase (“1”). The mismatch of the shrinkage strain thus induces the shape transformation.”

Comment 2. It would be useful to list the voxel dimensions in the experimentally printed material. Looking at the scale bars in Figure 7, it appears to be roughly on the order of a 1x1x1 mm? Building on this comment, the 3D printing material is interesting and may be useful in some applications, however the evaporation of acrylate monomer may be undesirable in others. If the dimensions at this scale for the voxels, it would be helpful to briefly discuss the applicability of this method to other voxelated printing schemes, perhaps using materials of different thermal expansion to achieve similar effects.

Response: Thanks for your valuable suggestion. The AC plate has a dimension of 42 mm × 42 mm × 0.6 mm, with a printing voxel size of $(50 \mu\text{m})^3$, which is adjustable based on the printer. This is distinct from the design voxel size of AC plate, which is 2.8mm×2.8mm×0.3mm in experiments (the AC plate has 15×15×2 design voxels). In theory, our design approach is generally applicable across different length scales. To demonstrate this, we printed a smaller AC plate measuring 21 mm × 21 mm × 0.3 mm, with a printing voxel size of $(25 \mu\text{m})^3$, using the

same material system. As shown in the newly added **Supplementary Movie 5**, the plate achieves the target shape morphing.

Moreover, our design approach is generally applicable to any active composite (AC) structure that exhibits differential dimensional changes (or mismatched strains) under proper activation stimuli. To demonstrate this, we have designed two additional grayscale DLP printable material systems that utilize different actuation mechanisms. One system operates under the same heating conditions, while the other uses solvent to induce the shape morphing. As shown in the newly added **Supplementary Movie 6**, both systems achieve the target shape change, demonstrating the applicability not only across different material systems but also under varied activation stimuli.

The above results suggest the general applicability of our approach to various voxelated printing techniques that may use different materials (thus having different properties), activation stimuli, and on different length scales. We have added above results and discussions in our revised manuscript (see highlighted text on Pages 29-30, 36, 42):

Pages 29-30

For the same case, we further implement our design in a smaller AC plate halved in size. As shown in **Supplementary Movie 5**, the printed sheet morphs upon heating and eventually achieves the target, which validates our design on such a smaller length scale. Moreover, we print our optimal design using two additional material systems that employ distinct actuation mechanisms and successfully achieved the target shape change in both cases (see **Supplementary Movie 6** for the shape-morphing process and the actuated sheet). Details on the length scale and material systems are provided in “Methods”.

.....

Moreover, our design approach demonstrates general applicability across various material systems, actuation mechanisms, and length scales.

Page 36

In addition, our design approach is generally applicable across various material systems, actuation mechanisms, and length scales. This also implies the general applicability to other voxelated printing techniques.

Page 42

Alternatively, two additional grayscale DLP printable material systems were presented in **Supplementary Movie 8**. The first one, which undergoes deformation under the same heating conditions, consists of trimethylolpropane triacrylate (Sigma-Aldrich), Ebecryl 8402, and n-butyl acrylate (Sigma-Aldrich) in a weight ratio of 1:2:2, with the same loading of additives (1 wt% photoinitiator, 0.08 wt% photoabsorber, and 0.04 wt% fluorescent dye, the same for the second one). The second material system is composed of poly(ethylene glycol) diacrylate (Sigma-Aldrich) and 2-hydroxyethyl acrylate (Sigma-Aldrich) in a weight ratio of 1:1. Activation is achieved by swelling in acetone for approximately 7 minutes, followed by drying in air.

REVIEWERS' COMMENTS

Reviewer #2 (Remarks to the Author):

The reviewer still believes that the innovation of this paper overlaps greatly with previous research (DOI: 10.1002/adfm.202109805), and it is not enough for publication in this journal. Rejection is strongly recommended.

Reviewer #3 (Remarks to the Author):

The manuscript can be accepted now.

Reviewer #4 (Remarks to the Author):

The authors' responses to the reviewer comments are satisfactory.

RESPONSE TO REVIEWERS

Reviewer #2 (Remarks to the Author):

The reviewer still believes that the innovation of this paper overlaps greatly with previous research (DOI: 10.1002/adfm.202109805), and it is not enough for publication in this journal. Rejection is strongly recommended.

Response: We respectfully disagree with your assessment regarding the significance of this work compared to our previous research (Ref.40, AFM, 2022). As detailed in our response from the last round, our current work introduces significant advancements in the field of 4D printing by addressing inverse designs for 3D shape morphing of plates, considering that (1) the transition from 2D to 3D introduces significant new challenges, (2) we develop novel strategies to tackle the challenges, and (3) the current work has a broader range of applications in 4D printing.

Reviewer #3 (Remarks to the Author):

The manuscript can be accepted now.

Reviewer #4 (Remarks to the Author):

The authors' responses to the reviewer comments are satisfactory.